# Geometrical aspects of lattice gauge equivariant convolutional neural networks

**David I. Müller**  *dmueller@hep.itp.tuwien.ac.at*
*TU Wien, Institute for Theoretical Physics, A-1040 Vienna, Austria*

**Jimmy Aronsson**  *jimmyar@chalmers.se*
*Chalmers University of Technology, Department of Mathematical Sciences,*
*SE-412 96 Gothenburg, Sweden*

**Daniel Schuh**  *schuh@hep.itp.tuwien.ac.at*
*TU Wien, Institute for Theoretical Physics, A-1040 Vienna, Austria*

**Reviewed on OpenReview:** *https://openreview.net/forum?id=zO4aAVHxPe*

## Abstract

Lattice gauge equivariant convolutional neural networks (L-CNNs) are a framework for convolutional neural networks that can be applied to non-abelian lattice gauge theories without violating gauge symmetry. We demonstrate how L-CNNs can be equipped with global group equivariance. This allows us to extend the formulation to be equivariant not just under translations but under global lattice symmetries such as rotations and reflections. Additionally, we provide a geometric formulation of L-CNNs and show how convolutions in L-CNNs arise as a special case of gauge equivariant neural networks on $\mathrm{SU}(N)$ principal bundles.

## 1 Introduction

In recent years, machine learning methods incorporating ideas based on symmetry and geometry, often summarized under the term geometric deep learning (Bronstein et al., 2021; Gerken et al., 2023), have received much attention in both computer vision and physics. Most famously, convolutional neural networks (CNNs) (LeCun et al., 1989) have proven to be an excellent machine learning architecture for computer vision tasks such as object detection and classification. The classic examples include determining whether an image contains a particular animal (e.g. cat or dog (Parkhi et al., 2012)), or identifying numbers in images of hand-written digits (Deng, 2012). For these tasks, it has been demonstrated that CNN architectures excel both in terms of accuracy and reduced model size, i.e. the number of model parameters. A key differentiating feature of CNNs compared to generic neural networks is that they are formulated as stacks of convolutional layers, which exhibit translational symmetry or, more accurately, translational equivariance. If a translation is applied to the input of a convolutional layer, then the resulting output will be appropriately shifted as well. This equivariance property is highly useful in the case of image classification, where the absolute position of a particular feature (a cat; a hand-written digit) in the image is not important. Translational equivariance further implies weight sharing, which reduces the number of required model parameters and training time. Consequently, CNNs provide not just more accurate but also more robust models compared to their translationally non-symmetric counterparts.

Whereas CNNs are only guaranteed to be translation equivariant, the concept of equivariance in neural networks can be extended to symmetries beyond translations, such as rotations or reflections. Group equivariant CNNs (G-CNNs) (Cohen & Welling, 2016; Cohen et al., 2019a; Aronsson, 2022) and steerable CNNs (Cohen & Welling, 2017; Weiler et al., 2018; Cesa et al., 2022) use convolutions on groups to achieve equivariance with respect to general global symmetries. Analogous to the equivariance property of traditional CNNs, group

transformations (e.g. roto-translations) applied to the input of a group equivariant convolutional layer, lead to the same group transformation being consistently applied to the output. Group convolutional layers thus commute with group transformations. In certain applications where larger symmetries are important, these networks have been shown to further improve performance compared to networks exhibiting less symmetry (Graham et al., 2020; Gerken et al., 2022). From a physical perspective, the symmetries considered in CNNs and, more generally, in G-CNNs are analogous to global symmetries of lattice field theories, which has led to numerous applications of CNNs in high energy physics (see Boyda et al. (2022) for a review). For example, CNNs have been applied to detect phase transitions and learn observables (Zhou et al., 2019; Blücher et al., 2020; Bachtis et al., 2020; Bulusu et al., 2021; Bachtis et al., 2021) and as generative models (Nicoli et al., 2021; Albergo et al., 2021a; de Haan et al., 2021; Gerdes et al., 2023; Albergo et al., 2021b) in both scalar and fermionic lattice field theories.

In addition to global symmetries, the laws of physics of the fundamental interactions are based on the notion of local symmetry, which is the foundation of gauge theories. Local symmetries allow for group transformations that can differ at every point in space-time. In machine learning, gauge equivariant neural networks (Cohen et al., 2019b; Cheng et al., 2019) (see also Gerken et al. (2023) for a review) have been proposed as architectures that are well-suited for data living on curved manifolds. In high energy physics, similar methods have been applied to problems in lattice gauge theory. For example, gauge symmetric machine learning models have been used as generative models (Kanwar et al., 2020; Boyda et al., 2021; Albergo et al., 2021a; Abbott et al., 2022; Bacchio et al., 2023) to avoid the problem of critical slowing down inherent to Markov Chain Monte Carlo simulations at large lattice sizes, or as machine-learned preconditioners for the Dirac equation in lattice QCD (Lehner & Wettig, 2023a;b; Knüttel et al., 2024). Going beyond specific applications, Lattice gauge equivariant CNNs (L-CNNs) (Favoni et al., 2022) have recently been proposed as a general gauge equivariant architecture for generic machine learning problems in lattice gauge theory. L-CNNs use gauge field configurations as input and can process data in a manner compatible with gauge symmetry. They consist of a set of gauge equivariant layers to build up networks as stacks of individual layers. In particular, gauge equivariant convolutional layers (L-Convs) are convolutional layers which use parallel transport to preserve gauge symmetry while combining data at different lattice sites. Because of their expressiveness, L-CNNs can be used as universal approximators of arbitrary gauge equivariant and invariant functions. It has been demonstrated in Favoni et al. (2022) that L-CNNs can accurately learn gauge invariant observables such as Wilson loops from datasets of gauge field configurations. Similar to CNNs, the convolutions used in L-CNNs are equivariant under lattice translations.

In this paper, we revisit L-CNNs from a geometric point of view and extend them by including a larger degree of lattice symmetry. First, we review lattice gauge theory and the original formulation of L-CNNs in Section 2. L-CNNs were originally constructed by incorporating local symmetry into ordinary CNNs, which means that L-CNNs are equivariant under lattice translations but not under other lattice symmetries such as rotations and reflections. We remedy this in Section 3 by applying methods from G-CNNs to L-CNNs. Our main result is a gauge equivariant convolution that can be applied to tensor fields and that is equivariant under translations, rotations, and reflections. Finally, in Section 4, we put the original L-CNNs in a broader context by relating them to a mathematical theory for equivariant neural networks. In doing so, we demonstrate how convolutions in L-CNNs can be understood as discretizations of convolutions on $SU(N)$ principal bundles.

## 2 Theoretical background

In this section, we review Yang-Mills theory and lattice gauge theory in the way these topics are usually introduced within high-energy physics, largely following the conventions of Peskin & Schroeder (1995) and Gattringer & Lang (2010). Having defined concepts such as gauge symmetry and gauge invariance, we then review aspects of L-CNNs.

### 2.1 Yang-Mills theory

We consider $SU(N)$ Yang-Mills theory on Euclidean space-time $\mathcal{M} = \mathbb{R}^D$ with $D - 1 > 0$ spatial dimensions. We choose Cartesian coordinates $x^\mu$ on $\mathcal{M}$ with $\mu \in \{1, 2, \ldots, D\}$ such that the metric on $\mathcal{M}$ is Eu-

clidean, i.e. $g_{\mu\nu} = \delta_{\mu\nu}$, where $\delta_{\mu\nu}$ is the Kronecker symbol. The degrees of freedom in this theory are gauge fields $A_\mu(\mathbf{x})$, which are $\mathfrak{su}(N)$-valued vector fields on $\mathcal{M}$. We further choose a matrix representation of $\mathfrak{su}(N)$, namely the fundamental representation spanned by the generators $t^a \in \mathbb{C}^{N \times N}$ with $a \in \{1, 2, \ldots, N^2 - 1\}$, which are traceless Hermitian matrices usually normalized to satisfy

$$\mathrm{Tr}\left[t^a t^b\right] = \frac{1}{2}\delta^{ab}. \tag{1}$$

We note that the convention to use traceless Hermitian matrices to describe $\mathfrak{su}(N)$, in contrast to anti-Hermitian matrices, is often used in high-energy physics. These two conventions are related by replacing $t^a \to it^a$, where $i$ is the imaginary unit. With a basis for both $\mathcal{M}$ and $\mathfrak{su}(N)$, a gauge field can be written as the 1-form

$$\mathcal{A}(\mathbf{x}) = A_\mu(\mathbf{x})\mathrm{d}x^\mu = A_\mu^a(\mathbf{x})t^a\mathrm{d}x^\mu, \tag{2}$$

with components $A_\mu^a : \mathcal{M} \to \mathbb{R}$. Two different gauge fields $\mathcal{A}$ and $\mathcal{A}'$ are considered to be gauge equivalent if their components can be related via a gauge transformation $T_\Omega$,

$$A_\mu'(\mathbf{x}) = T_\Omega A_\mu(\mathbf{x}) = \Omega(\mathbf{x})(A_\mu(\mathbf{x}) - i\partial_\mu)\Omega^\dagger(\mathbf{x}), \tag{3}$$

where $\Omega : \mathcal{M} \to \mathrm{SU}(N)$ is a differentiable function on space-time. Gauge fields that can be related via gauge transformations form an equivalence class. Within gauge theory, (the components of) gauge fields are not considered physical, observable fields. Rather, the physical state of a system is the same for all gauge fields in any particular equivalence class. Observables within gauge theory therefore must be gauge invariant functionals of $\mathcal{A}$. The most prominent example of such a gauge invariant functional is the Yang-Mills action

$$S[\mathcal{A}] = \frac{1}{2g^2} \int_{\mathcal{M}} \mathrm{d}^D x \, \mathrm{Tr}\left[F_{\mu\nu}(\mathbf{x})F^{\mu\nu}(\mathbf{x})\right], \tag{4}$$

which maps a gauge field $\mathcal{A}$ to a single real number $S[\mathcal{A}] \in \mathbb{R}$. Here, $g > 0$ is the Yang-Mills coupling constant, and the $\mathfrak{su}(N)$-valued field strength tensor is given by

$$F_{\mu\nu}(\mathbf{x}) = \partial_\mu A_\nu(\mathbf{x}) - \partial_\nu A_\mu(\mathbf{x}) + i\left[A_\mu(\mathbf{x}), A_\nu(\mathbf{x})\right], \tag{5}$$

where $[\,,\,]$ denotes the commutator of matrices in the fundamental representation of $\mathfrak{su}(N)$. Under gauge transformations, the field strength tensor is transformed according to

$$T_\Omega F_{\mu\nu}(\mathbf{x}) = \Omega(\mathbf{x})F_{\mu\nu}(\mathbf{x})\Omega^\dagger(\mathbf{x}). \tag{6}$$

Because of the transformation behavior of the field strength tensor and the trace in the Yang-Mills action, the value of the action is invariant under gauge transformations, i.e.

$$S[T_\Omega \mathcal{A}] = S[\mathcal{A}]. \tag{7}$$

The invariance of the Yang-Mills action under gauge transformations is called gauge symmetry.

## 2.2 Lattice gauge theory

Lattice discretizations of non-abelian Yang-Mills theory with exact lattice gauge symmetry can be constructed with the help of the link formalism of lattice gauge theory (Wilson, 1974). In this formalism, the gauge fields $A_\mu(\mathbf{x}) \in \mathfrak{su}(N)$ are replaced by gauge link variables $U_{\mathbf{x},\mu} \in \mathrm{SU}(N)$ defined on the edges of a finite hypercubic lattice $\Lambda$ with periodic boundary conditions. We use the fundamental representation of $\mathfrak{su}(N)$ and $\mathrm{SU}(N)$ to represent gauge fields and gauge links as complex matrices. The links $U_{\mathbf{x},\mu}$ connect a lattice site $\mathbf{x}$ to its neighboring sites $\mathbf{x} + \mu = \mathbf{x} + a\hat{e}_\mu$, separated by the lattice spacing $a$ and the Euclidean basis vector $\hat{e}_\mu$. The inverse link, which connects $\mathbf{x} + \mu$ to $\mathbf{x}$, is written as $U_{\mathbf{x}+\mu,-\mu} = U_{\mathbf{x},\mu}^\dagger$.

In terms of the gauge field, a gauge link is given by the path-ordered exponential

$$U_{\mathbf{x},\mu} = \mathcal{P} \exp\left\{i \int_0^1 \mathrm{d}s \frac{\mathrm{d}x^\nu(s)}{\mathrm{d}s} A_\nu(\mathbf{x}(s))\right\}. \tag{8}$$

In the convention that is used here, the path ordering operator $\mathcal{P}$ shifts fields earlier in the path to the left and fields later to the right of the product. The function $\mathbf{x}(s) : [0, 1] \rightarrow \mathbb{R}^D$ parameterizes the straight-line path connecting $\mathbf{x}$ to $\mathbf{x} + \mu$. Geometrically, the gauge links prescribe how to parallel transport along the edges of the lattice. They transform under general lattice gauge transformations $T_\Omega$, $\Omega : \Lambda \rightarrow \mathrm{SU}(N)$ according to

$$T_\Omega U_{\mathbf{x},\mu} = \Omega_{\mathbf{x}} U_{\mathbf{x},\mu} \Omega_{\mathbf{x}+\mu}^\dagger. \tag{9}$$

Gauge links are the shortest possible Wilson lines on the lattice. Longer Wilson lines are formed by multiplying links that connect consecutive points to form an arbitrary path on the lattice. For closed paths, they are referred to as Wilson loops, and the smallest loop, which is the $1 \times 1$ loop, is called a plaquette and reads

$$U_{\mathbf{x},\mu\nu} = U_{\mathbf{x},\mu} U_{\mathbf{x}+\mu,\nu} U_{\mathbf{x}+\mu+\nu,-\mu} U_{\mathbf{x}+\nu,-\nu}. \tag{10}$$

It transforms under gauge transformations as given by

$$T_\Omega U_{\mathbf{x},\mu\nu} = \Omega_{\mathbf{x}} U_{\mathbf{x},\mu\nu} \Omega_{\mathbf{x}}^\dagger. \tag{11}$$

The Wilson action (Wilson, 1974), which can be written in terms of plaquettes, reads

$$S_W[U] = \frac{2}{g^2} \sum_{\mathbf{x} \in \Lambda} \sum_{\mu < \nu} \mathrm{Re} \, \mathrm{Tr} \left[ \mathbb{1} - U_{\mathbf{x},\mu\nu} \right]. \tag{12}$$

Note that (12) is invariant under global symmetries of the lattice such as translations, discrete rotations, and reflections. Its invariance under lattice gauge transformations follows from the local transformation property of the plaquettes and the trace.

In the continuum limit, for small lattice spacings $a \ll 1$, gauge links can be approximated by the matrix exponential

$$U_{\mathbf{x},\mu} \approx \exp\left( i a A_\mu \left( \mathbf{x} + \frac{1}{2}\mu \right) \right) \tag{13}$$

at the midpoint $\mathbf{x} + \frac{1}{2}\mu$. Furthermore, in this limit, plaquettes approximate the non-abelian field strength tensor, given by Eq. (5),

$$U_{\mathbf{x},\mu\nu} \approx \exp\left( i a^2 F_{\mu\nu} \left( \mathbf{x} + \frac{1}{2}\mu + \frac{1}{2}\nu \right) \right), \tag{14}$$

and the Wilson action approximates the Yang-Mills action, introduced in Eq. (4).

### 2.3 Lattice gauge equivariant convolutional neural networks

An L-CNN is built up by individual layers $\Phi$, which take as input at least one tuple $(\mathcal{U}, \mathcal{W})$. The first part of the tuple, $\mathcal{U} = \{U_{\mathbf{x},\mu}\}$, is a set of gauge links in the fundamental representation that transform non-locally as in Eq. (9). The second part, $\mathcal{W} = \{W_{\mathbf{x},a}\}$, is a set of complex matrices $W_{\mathbf{x},a} \in \mathbb{C}^{N \times N}$ that transform locally, like plaquettes, as in Eq. (11):

$$T_\Omega W_{\mathbf{x},a} = \Omega_{\mathbf{x}} W_{\mathbf{x},a} \Omega_{\mathbf{x}}^\dagger. \tag{15}$$

Here, the index $a \in \{1, \ldots N_{\mathrm{ch}}\}$ refers to the channel, and $N_{\mathrm{ch}}$ denotes the total number of channels in the layer in question. The output of the layer is, generally, again a tuple $(\mathcal{U}', \mathcal{W}')$, with possibly a different number of channels. We require every layer to be lattice gauge equivariant in the sense of

$$\Phi(T_\Omega \mathcal{U}, T_\Omega \mathcal{W}) = T_\Omega' \Phi(\mathcal{U}, \mathcal{W}), \tag{16}$$

where $T_\Omega'$ denotes the application of the gauge transformation $\Omega$ to the output of the layer. Generally, one can consider layers where $T_\Omega' \neq T_\Omega$. This would be the case if the representation of the input tuple $(\mathcal{U}, \mathcal{W})$ is different from the one of the output tuple $(\mathcal{U}', \mathcal{W}')$. In this work, we only consider layers that do not change the representation of $\mathrm{SU}(N)$ or the transformation behavior of the links $\mathcal{U}$ and the locally transforming matrices $\mathcal{W}$ in any way. Additionally, we only focus on layers that do not modify the set of gauge links $\mathcal{U}$, i.e. we always require that $\mathcal{U}' = \mathcal{U}$. A network built from multiple gauge equivariant layers $\{\Phi_1, \Phi_2, \ldots, \Phi_N\}$

through composition, $\Phi_N \circ \cdots \circ \Phi_2 \circ \Phi_1$, respects lattice gauge equivariance in the sense of Eq. (16). In the following, we review some of the layers introduced in Favoni et al. (2022).

A convolutional layer usually aims to combine data from different locations with trainable weights in a translationally equivariant manner. In an L-CNN, such a layer is required to respect lattice gauge equivariance as well. This is fulfilled by the Lattice gauge equivariant Convolutional (L-Conv) layer, which is a map $(\mathcal{U}, \mathcal{W}) \mapsto (\mathcal{U}, \mathcal{W}')$, defined as

$$W'_{\mathbf{x},a} = \sum_{b,\mu,k} \psi_{a,b,\mu,k} U_{\mathbf{x},k\cdot\mu} W_{\mathbf{x}+k\cdot\mu,b} U^\dagger_{\mathbf{x},k\cdot\mu}, \tag{17}$$

with the trainable weights $\psi_{a,b,\mu,k} \in \mathbb{C}$, output channel index $1 \leq a \leq N_{\mathrm{ch,out}}$, input channel index $1 \leq b \leq N_{\mathrm{ch,in}}$, lattice directions $1 \leq \mu \leq D$ and distances $-N_k \leq k \leq N_k$, where $N_k$ determines the kernel size. Note that the output channels associated with $W'$ are a linear combination of the input channels of $W$. In general, the number of input channels $N_{\mathrm{ch,in}}$ may differ from the number of output channels $N_{\mathrm{ch,out}}$. The matrices $U_{\mathbf{x},k\cdot\mu}$ appearing in Eq. (17) describe parallel transport starting at the point $\mathbf{x}$ to the point $\mathbf{x} + k \cdot \mu$. They are given by

$$U_{\mathbf{x},k\cdot\mu} = \prod_{i=0}^{k-1} U_{\mathbf{x}+i\cdot\mu,\mu} = U_{\mathbf{x},\mu} U_{\mathbf{x}+\mu,\mu} U_{\mathbf{x}+2\cdot\mu,\mu} \ldots U_{\mathbf{x}+(k-1)\cdot\mu,\mu} \tag{18}$$

for positive $k$ and

$$U_{\mathbf{x},k\cdot\mu} = \prod_{i=0}^{k-1} U_{\mathbf{x}-i\cdot\mu,-\mu} = U_{\mathbf{x},-\mu} U_{\mathbf{x}-\mu,-\mu} U_{\mathbf{x}-2\cdot\mu,-\mu} \ldots U_{\mathbf{x}-(k-1)\cdot\mu,-\mu} \tag{19}$$

for negative $k$. Only parallel transports along straight paths are considered because the shortest path between two lattice sites is not unique otherwise. Data on lattice points that are not connected by straight paths can be combined by stacking multiple layers. A bias term can be included by adding the unit element $\mathbb{1}$ to the set $\mathcal{W}$. A further increase in expressivity can be achieved by also adding the Hermitian conjugates of $W_{\mathbf{x},i}$ to $\mathcal{W}$. A general L-Conv layer thus may be written as

$$W'_{\mathbf{x},a} = \sum_{b,\mu,k} \psi_{a,b,\mu,k} U_{\mathbf{x},k\cdot\mu} W_{\mathbf{x}+k\cdot\mu,b} U^\dagger_{\mathbf{x},k\cdot\mu} + \sum_{b,\mu,k} \tilde{\psi}_{a,b,\mu,k} U_{\mathbf{x},k\cdot\mu} W^\dagger_{x+k\cdot\mu,b} U^\dagger_{\mathbf{x},k\cdot\mu} + \psi_0 \mathbb{1}, \tag{20}$$

with weights $\psi_{a,b,\mu,k}$, $\tilde{\psi}_{a,b,\mu,k}$ and a bias term $\psi_0$. For brevity, we will use the more compact form given in Eq. (17). L-Conv layers are gauge equivariant by virtue of the transformation behavior of the parallel transporters $U_{\mathbf{x},k\cdot\mu}$. From Eq. (9) it follows that

$$T_\Omega U_{\mathbf{x},k\cdot\mu} = \Omega_{\mathbf{x}} U_{\mathbf{x},k\cdot\mu} \Omega^\dagger_{\mathbf{x}+k\cdot\mu}. \tag{21}$$

The matrices $U_{\mathbf{x},k\cdot\mu}$ thus allow the L-Conv layer to combine data from various lattice sites without violating gauge symmetry.

It follows that if data are combined only locally, there is no need for parallel transport to construct a lattice gauge equivariant layer. This is realized by Lattice gauge equivariant Bilinear (L-Bilin) layers, which are maps $(\mathcal{U}, \mathcal{W}), (\mathcal{U}, \mathcal{W}') \mapsto (\mathcal{U}, \mathcal{W}'')$, given by

$$W''_{\mathbf{x},a} = \sum_{b,c} \alpha_{a,b,c} W_{\mathbf{x},b} W'_{\mathbf{x},c}. \tag{22}$$

The weights $\alpha_{a,b,c} \in \mathbb{C}$ are trainable, have an output channel index $1 \leq a \leq N_{\mathrm{ch,out}}$ and two input channel indices $1 \leq b \leq N_{\mathrm{ch,in}}$ and $1 \leq c \leq N'_{\mathrm{ch,in}}$. Analogously to the L-Conv layer, the unit element and the Hermitian conjugates can be added to $\mathcal{W}$ to increase the expressivity of the L-Bilin layer.

Lattice gauge equivariant Activation functions (L-Act), which are maps $(\mathcal{U}, \mathcal{W}) \mapsto (\mathcal{U}, \mathcal{W}')$, are the generalization of standard activation functions to the L-CNN. They can be applied at every lattice site and are given by

$$W'_{\mathbf{x},a} = \nu_{\mathbf{x},a}(\mathcal{W}) W_{\mathbf{x},a}, \tag{23}$$

where $\nu$ is any scalar-valued and gauge invariant function. One option is $\nu_{\mathbf{x},a}(\mathcal{W}) = \Theta(\mathrm{Re}(\mathrm{Tr}(W_{\mathbf{x},a})))$, with the Heaviside function $\Theta$. For real-valued scalars $s$, this would lead to the well-known ReLU activation function, which can also be written as $\mathrm{ReLU}(s) = \Theta(s)s$.

Finally, the last important layer to consider is the Trace layer. This layer maps $(\mathcal{U}, \mathcal{W}) \mapsto \mathcal{T}_{\mathbf{x},a}$, which converts the lattice gauge equivariant quantities $(\mathcal{U}, \mathcal{W})$ to lattice gauge invariant quantities

$$\mathcal{T}_{\mathbf{x},a}(\mathcal{U}, \mathcal{W}) = \mathrm{Tr}(W_{\mathbf{x},a}). \tag{24}$$

A gauge invariant layer such as this is necessary if the network output is supposed to approximate a (gauge invariant) physical observable.

As an example, an L-CNN can compute the plaquettes as a pre-processing step and use them as local variables $W_{\mathbf{x},a}$ in the subsequent layers. With stacks of L-Conv and L-Bilin layers, it can build arbitrarily shaped loops, depending on the number of these stacks (Favoni et al., 2022). The network expressivity can be increased further by introducing L-Acts between said stacks, and if the output is a physical observable, there should be a Trace layer at the end. After the Trace layer, a conventional CNN or other neural network can be added without breaking lattice gauge equivariance.

## 3 Extending L-CNNs to general group equivariance

In the last section, we have reviewed the L-Conv operation on a hypercubic lattice $\Lambda = \mathbb{Z}^D$. Like a standard convolutional layer, the L-Conv layer is equivariant under (integer) translations on $\mathbb{Z}^D$, which, when interpreted as a group $\mathbb{T}$, can be identified with the lattice itself, $\mathbb{T} \sim \mathbb{Z}^D$. However, lattice gauge theories on hypercubic lattices typically exhibit larger isometry groups $G$ that include discrete rotations and reflections, which we denote by the subgroup $K \subset G$. In this section, we explicitly construct L-CNN layers that are compatible with general $G$-symmetry.

The original approach to G-CNNs (Cohen & Welling, 2016) is based on promoting feature maps from functions on the lattice $\mathbb{Z}^D$, or, more generally, functions on some space $\mathcal{M}$, to functions on the group $G$. Group equivariant convolutions, or $G$-convolutions, are analogous to traditional convolutions, except that integrals (or sums in the case of a discrete space $\mathcal{M}$) are carried out over the whole group $G$. In contrast, the modern approach to G-CNNs on homogeneous spaces $\mathcal{M}$ uses a fiber bundle formalism (Cohen et al., 2019a; Aronsson, 2022) in which feature maps are modeled via fields on $\mathcal{M}$, that is, via sections of associated vector bundles over $\mathcal{M}$. This approach is geometrically pleasing because it means that the inputs to and outputs from convolutional layers live directly on $\mathcal{M}$. It also offers computational advantages since convolutional layers become integrals over $\mathcal{M} \simeq G/K$ rather than integrals over the larger space $G$. Here, $K$ is the subgroup of $G$ that stabilizes an arbitrarily chosen origin in $\mathcal{M}$.

However, in order to take advantage of this simplification, one needs to identify feature maps $f : G \to \mathbb{R}^N$ with fields on $\mathcal{M}$. This imposes a constraint given by

$$f(gk) = \rho(k)^{-1}f(g), \tag{25}$$

where $g \in G, k \in K$, and $\rho$ is a representation of $K$ (see Aronsson (2022) for details). This constraint is difficult to enforce numerically and is sometimes ignored, preventing the geometric view of $f$ as a field on $\mathcal{M}$. Fortunately, ignoring the constraint effectively promotes $f$ to a field on the group $G$ instead, similar to the approach laid out in Cohen & Welling (2016), and the bundle formalism still applies after changing the homogeneous space from $\mathcal{M}$ to $G$. Even though ignoring the constraint makes convolutional layers more expensive to compute, it drastically simplifies their implementation in machine learning frameworks such as *PyTorch* (Paszke et al., 2019). In our case, this is because the group $G$ of lattice symmetries is a semi-direct product $G = \mathbb{T} \rtimes K$ of translations $x \in \mathbb{T}$ and rotoreflections $r \in G/\mathbb{T} = K$. Group elements can thus be split into products $g = xr$. Consequently, feature maps $f : G \to \mathbb{R}^N$ can be viewed as "stacks" of feature maps $f_r(x)$ on the lattice $\mathbb{Z}^D$. It can be shown that $G$-convolutions can be expressed in terms of traditional $\mathbb{Z}^D$-convolutions (Cohen & Welling, 2016), for which highly efficient implementations already exist.

Our strategy to develop a $G$-equivariant framework for L-CNNs is thus the following: We first review group equivariant networks without gauge symmetry by working out explicit $G$-convolutions for scalar fields, vector

fields, and general tensor fields discretized on the lattice $\mathbb{Z}^D$, in the spirit of the original G-CNN formulation (Cohen & Welling, 2016). We then show how $G$-convolutions can be combined with our approach to lattice gauge equivariant convolutions to obtain fully $G$-equivariant L-Convs. We extend our approach to bilinear layers (L-Bilin), activation layers (L-Act), trace layers, and pooling layers, which allows us to formulate fully $G$-equivariant L-CNNs.

### 3.1 Group equivariant convolutions for scalars on the lattice

Convolutional layers in traditional CNNs act on feature maps, e.g. functions $f : \mathbb{Z}^D \to \mathbb{R}^n$, where $n$ is the number of channels. For explicitness, we consider real-valued feature maps. A real-valued convolution with $n$ input channels and $n'$ output channels is given by

$$[\psi * f]^a(\mathbf{x}) = \sum_{b=1}^{n} \sum_{\mathbf{y} \in \mathbb{Z}^D} \psi^{ab}(\mathbf{y} - \mathbf{x}) f^b(\mathbf{y}), \quad a \in \{1, 2, \dots n'\}, \tag{26}$$

where $\psi : \mathbb{Z}^D \to \mathbb{R}^{n' \times n}$ are the kernel weights. Here we explicitly use boldfaced letters to denote points on the lattice $\mathbb{Z}^D$ and the symbol $*$ to denote a $\mathbb{Z}^D$-convolution. The convolution operation is equivariant with respect to translations: applying a translation $z \in \mathbb{T}$, which can be identified with the point $\mathbf{z} \in \mathbb{Z}^D$, to $[\psi * f]$ yields

$$\begin{aligned} L_z[\psi * f]^a(\mathbf{x}) &= [\psi * f]^a(\mathbf{x} - \mathbf{z}) \\ &= \sum_{b=1}^{n} \sum_{\mathbf{y} \in \mathbb{Z}^D} \psi^{ab}(\mathbf{y} - \mathbf{x} + \mathbf{z}) f^b(\mathbf{y}) \\ &= \sum_{b=1}^{n} \sum_{\mathbf{y}' \in \mathbb{Z}^D} \psi^{ab}(\mathbf{y}' - \mathbf{x}) f^b(\mathbf{y}' - \mathbf{z}) \\ &= [\psi * L_z f]^a(\mathbf{x}). \end{aligned} \tag{27}$$

The left translation $L_z$ commutes with the convolution operation. The main idea of Cohen & Welling (2016) is to introduce convolutions that are $G$-equivariant, i.e. that commute with $L_g$ for $g \in G$. More specifically, two types of $G$-convolutions ($G$-Convs) are introduced: the first-layer $G$-convolution acts on feature maps on $\mathbb{Z}^D$ and promotes them to feature maps on the group $G$, and the full $G$-convolution, which acts on feature maps on $G$ and outputs feature maps on $G$. To simplify notation and without loss of generality, we set the number of channels to one.

The first-layer $G$-convolution acting on a feature map $f : \mathbb{Z}^D \to \mathbb{R}$ is given by (Cohen & Welling, 2016)

$$[\psi \star f](g) = \sum_{\mathbf{y} \in \mathbb{Z}^D} \psi(g^{-1} \cdot \mathbf{y}) f(\mathbf{y}), \quad g \in G, \tag{28}$$

where $\psi : \mathbb{Z}^D \to \mathbb{R}$ are the kernel weights, and $g^{-1} \cdot \mathbf{y}$ denotes the action of the group element $g^{-1}$ on the point $\mathbf{y} \in \mathbb{Z}^D$. Note that we denote $\mathbb{Z}^D$-convolutions, as in (26), by $*$ and $G$-convolutions by $\star$. Uniquely splitting the group element $g$ into a translation $x \in \mathbb{T}$ (identified with $\mathbf{x} \in \mathbb{Z}^D$) and a rotoreflection $r \in K = G/\mathbb{T}$ about the origin, $g = xr$, we have

$$g^{-1} \cdot \mathbf{y} = R^{-1}(\mathbf{y} - \mathbf{x}), \tag{29}$$

where $R \in \mathbb{Z}^{D \times D}$ is a matrix representation of the rotoreflection $r$. This split is unique because $G$ is a semi-direct product $G = \mathbb{T} \rtimes K$. In addition, the translations $\mathbb{T}$ form a normal subgroup of $G$:

$$g^{-1} x g \in \mathbb{T}, \quad \forall x \in \mathbb{T}, g \in G. \tag{30}$$

Note that the result of the $G$-convolution in Eq. (28) is a function $[\psi \star f] : G \to \mathbb{R}$ on the group $G$. The effect of the rotoreflection $r$ is that the feature map $f$ is convolved with the rotated kernel

$$L_r \psi(\mathbf{y} - \mathbf{x}) = \psi(R^{-1}(\mathbf{y} - \mathbf{x})). \tag{31}$$

The first-layer $G$-Conv can therefore be written as a convolution over $\mathbb{Z}^D$

$$[\psi \star f](xr) = [L_r \psi * f](\mathbf{x}), \tag{32}$$

which we refer to as the split form (see also section 7 of Cohen & Welling (2016)).

The full $G$-convolution acts on feature maps $f : G \to \mathbb{R}$ and is used after the first-layer $G$-convolution. It is given by

$$[\psi \star f](g) = \sum_{h \in G} \psi(g^{-1}h) f(h), \tag{33}$$

where $\psi : G \to \mathbb{R}$ are the kernel weights. Since we are dealing with discrete groups, we use a sum over the group elements to define the convolution. Both $g$ and $h$ are elements of the group $G$, and $g^{-1}h$ denotes the group product. Just as before, we would like to write this operation in terms of $\mathbb{Z}^D$-convolutions (the split form) using $g = xr$ and $h = ys$ with $x, y \in \mathbb{T}$ and $r, s \in G/\mathbb{T}$. In order to perform this split, we need to be able to interpret functions on $G$ as "stacks" of functions on $\mathbb{Z}^D$. Given $f : G \to \mathbb{R}$ and $h = ys$ we write

$$f(h) = f(ys) = f_s(\mathbf{y}), \tag{34}$$

where $f_s : \mathbb{Z}^D \to \mathbb{R}$ for each element $s \in G/\mathbb{T}$. The function $f$ is therefore equivalent to a stack of functions $\{f_s \mid s \in G/\mathbb{T}\}$. A left translation acting on $f$ with $g = xr$ and $h = ys$ induces

$$\begin{aligned}
L_g f(h) &= f(g^{-1}h) \\
&= f((xr)^{-1}ys) \\
&= f(r^{-1}x^{-1}yrr^{-1}s) \\
&= f_{r^{-1}s}(R^{-1}(\mathbf{y} - \mathbf{x})),
\end{aligned} \tag{35}$$

where $z = r^{-1}x^{-1}yr \in \mathbb{T}$ (as $\mathbb{T}$ is a normal subgroup of $G$) and $r^{-1}s \in G/\mathbb{T}$. In the last line we have made use of the fact that pure translations $x \in \mathbb{T}$ can be uniquely identified with points $\mathbf{x} \in \mathbb{Z}^D$ via the action of the translation subgroup on the origin $\mathbf{0}$. The point $\mathbf{z}$ associated with $z$ is given by

$$\mathbf{z} = z \cdot \mathbf{0} = r^{-1} \cdot ((x^{-1}y) \cdot (r \cdot \mathbf{0})) = r^{-1} \cdot ((x^{-1}y) \cdot \mathbf{0}) = r^{-1} \cdot (\mathbf{y} - \mathbf{x}) = R^{-1}(\mathbf{y} - \mathbf{x}), \tag{36}$$

where $r \cdot \mathbf{0} = \mathbf{0}$ because rotoreflections form the stabilizer subgroup associated with the origin.

The kernel in Eq. (33) can thereby be written as

$$\psi(g^{-1}h) = \psi_{r^{-1}s}(R^{-1}(\mathbf{y} - \mathbf{x})), \tag{37}$$

hence the split form of the full $G$-convolution is given by

$$\begin{aligned}
[\psi \star f](xr) &= \sum_{s \in G/\mathbb{T}} \sum_{\mathbf{y} \in \mathbb{Z}^D} \psi_{r^{-1}s}(R^{-1}(\mathbf{y} - \mathbf{x})) f_s(\mathbf{y}) \\
&= \sum_{s \in G/\mathbb{T}} [(L_r \psi_{r^{-1}s}) * f_s](\mathbf{x}),
\end{aligned} \tag{38}$$

which is a sum of multiple $\mathbb{Z}^D$-convolutions with rotated kernels $L_r \psi_{r^{-1}s}(\mathbf{y} - \mathbf{x})$. The split forms Eqs. (32) and (38) are particularly useful for concrete implementations in machine learning frameworks. Writing the $G$-convolutions in terms of $\mathbb{Z}^D$-convolutions allows us to make use of highly optimized implementations such as the *Conv2D* and *Conv3D* functions provided by *PyTorch*.

Both types of $G$-convolutions can be compactly written as

$$[\psi \star f](g) = \sum_{h \in H} L_g \psi(h) f(h), \tag{39}$$

where we use $H = \mathbb{Z}^D$ for the first-layer $G$-Conv and $H = G$ for the full $G$-Conv. In this form, it is evident that the two types merely differ in the group that is being summed over (translations $\mathbb{T} \sim \mathbb{Z}^D$ in the first-layer

$G$-Conv, the full group $G$ in the full $G$-Conv) and how the left translation acts on the kernel $\psi$. Depending on the choice of $H$, the left translated kernel $L_g\psi$ is either a rotated $\mathbb{Z}^D$-kernel for $H = \mathbb{Z}^D$ or a translated kernel on the group for $H = G$. It is now easy to check that $G$-convolutions are in fact equivariant under left translations $L_g$. Let $k \in G$, then we have

$$
\begin{aligned}
L_k[\psi \star f](g) &= [\psi \star f](k^{-1}g) \\
&= \sum_{h \in H} L_{k^{-1}g}\psi(h)f(h) \\
&= \sum_{h \in H} L_g\psi(kh)f(h) \\
&= \sum_{h' \in H} L_g\psi(h')f(k^{-1}h') \\
&= \sum_{h' \in H} L_g\psi(h')L_kf(h') \\
&= [\psi \star L_kf](g),
\end{aligned}
\tag{40}
$$

where we have used the substitution $h' = kh$ in the fourth line. For the first-layer $G$-Conv ($H = \mathbb{Z}^D$), where $h \in \mathbb{Z}^D$, $h' = kh$ is to be interpreted as a rotated and shifted coordinate on $\mathbb{Z}^D$ as in Eq. (29), whereas for the full $G$-Conv ($H = G$), $h' = kh$ is simply a translated group element in $G$. Note that $G$-equivariance can also be shown for the split forms Eqs. (32) and (38), but the proof is analogous to the one shown above.

### 3.2 Group equivariant convolutions for vector and tensor fields

We have explicitly shown that the $G$-convolutions are equivariant under general transformations $g$ via left translations $L_g$. The feature maps $f : \mathbb{Z}^D \to \mathbb{R}$, on which the convolutions act, transform like scalar fields under $L_g$ with $g = xr$:

$$
L_g f(\mathbf{y}) = f(R^{-1}(\mathbf{y} - \mathbf{x})).
\tag{41}
$$

In order to extend $G$-convolutions to vectors and tensors, we would expect transformations that act on the vector structure. For example, consider a vector field $v : \mathbb{Z}^D \to \mathbb{R}^D$ on the lattice with components $v^i : \mathbb{Z}^D \to \mathbb{R}$. Acting on $v$ with a general transformation $g = xr$ yields

$$
(L_g v)^i(\mathbf{y}) = R^i{}_j v^j(R^{-1}(\mathbf{y} - \mathbf{x})),
\tag{42}
$$

where $R^i{}_j$ are the components of the matrix representation of $r \in G/\mathbb{T}$ on $\mathbb{R}^D$. More generally, a type $(n,m)$ tensor field $w$, i.e. with $n$ vector and $m$ co-vector components, transforms according to

$$
(L_g w)^{i_1\dots i_n}{}_{j_1\dots j_m}(\mathbf{y}) = R^{i_1}{}_{i'_1}\dots R^{i_n}{}_{i'_n} w^{i'_1\dots i'_n}{}_{j'_1\dots j'_m}(R^{-1}(\mathbf{y} - \mathbf{x}))(R^{-1})^{j'_1}{}_{j_1}\dots(R^{-1})^{j'_m}{}_{j_m}.
\tag{43}
$$

Based on the compact form of scalar $G$-convolutions, Eq. (39), we now make the following guess at $G$-convolutions which map tensors of type $(n,0)$ to tensors of the same type:

$$
[\psi \star w]^{i_1\dots i_n}(g) = \sum_{h \in H} (L_g\psi)^{i_1\dots i_n}{}_{j_1\dots j_n}(h) w^{j_1\dots j_n}(h),
\tag{44}
$$

where $H = \mathbb{T} \sim \mathbb{Z}^D$ for the first-layer and $H = G$ for the full $G$-Conv. Here, the kernel $\psi$ is a tensor of type $(n,n)$ and acts as a general linear transformation of the tensor components of $w$. For the full $G$-Conv, $H = G$, left translations acting on tensor fields $\psi$ of type $(n,m)$ on the group are given by

$$
(L_g\psi)^{i_1\dots i_n}{}_{j_1\dots j_m}(h) = R^{i_1}{}_{i'_1}\dots R^{i_n}{}_{i'_n} \psi^{i'_1\dots i'_n}{}_{j'_1\dots j'_m}(g^{-1}h)(R^{-1})^{j'_1}{}_{j_1}\dots(R^{-1})^{j'_m}{}_{j_m}.
\tag{45}
$$

$G$-convolutions for tensors of mixed type $(n,m)$ are defined analogously.

The tensor $G$-convolutions can be cast into a more compact form by introducing condensed index notation: we write the block of indices $i_1 \ldots i_n$ as multi-indices $I$ so that

$$
\begin{aligned}
w^I{}_J &:= w^{i_1 \ldots i_n}{}_{j_1 \ldots j_n}, \\
\delta^I{}_J &:= \delta^{i_1}{}_{j_1} \ldots \delta^{i_n}{}_{j_n}, \\
R^I{}_J &:= R^{i_1}{}_{j_1} \ldots R^{i_n}{}_{j_n}.
\end{aligned}
\tag{46}
$$

Contractions of $R$ and $R^{-1}$ can be written as

$$
\begin{aligned}
R^I{}_J (R^{-1})^J{}_K &= R^{i_1}{}_{j_1} \ldots R^{i_n}{}_{j_n} (R^{-1})^{j_1}{}_{k_1} \ldots (R^{-1})^{j_n}{}_{k_n} \\
&= R^{i_1}{}_{j_1} (R^{-1})^{j_1}{}_{k_1} \ldots R^{i_n}{}_{j_n} (R^{-1})^{j_n}{}_{k_n} \\
&= \delta^{i_1}{}_{k_1} \ldots \delta^{i_n}{}_{k_n} \\
&= \delta^I{}_K.
\end{aligned}
\tag{47}
$$

Using multi-indices, the tensor $G$-convolutions are given by

$$
[\psi \star w]^I(g) = \sum_{h \in H} (L_g \psi)^I{}_J(h) w^J(h),
\tag{48}
$$

and left translations act on tensors according to

$$
(L_g w)^I{}_J(h) = R^I{}_{I'} w^{I'}{}_{J'}(g^{-1}h)(R^{-1})^{J'}{}_J.
\tag{49}
$$

Equivariance of Eq. (48) follows from

$$
\begin{aligned}
(L_k[\psi \star w])^I(g) &= R^I{}_{I'} [\psi \star w]^{I'}(k^{-1}g) \\
&= \sum_{h \in H} R^I{}_{I'} (L_{k^{-1}g} \psi)^{I'}{}_J(h) w^J(h) \\
&= \sum_{h \in H} R^I{}_{I'} (R^{-1})^{I'}{}_{I''} (L_g \psi)^{I''}{}_{J'}(kh) R^{J'}{}_J w^J(h) \\
&= \sum_{h' \in H} (L_g \psi)^I{}_J(h') R^{J'}{}_J w^J(k^{-1}h') \\
&= [\psi \star L_k w]^I(g),
\end{aligned}
\tag{50}
$$

where we have used the substitution $h' = kh$.

We can also define $G$-convolutions that change the tensor type. Given multi-indices $I_n = i_1 \ldots i_n$ and $J_m = j_1 \ldots j_m$ with $n \neq m$ in general, we define

$$
\tilde{w}^{I_n} = [\psi \star w]^{I_n}(g) = \sum_{h \in H} (L_g \psi)^{I_n}{}_{J_m}(h) w^{J_m}(h),
\tag{51}
$$

with $H = \mathbb{Z}^D$ for the first-layer and $H = G$ for the full $G$-Conv. Here, the output feature map $\tilde{w}$ is a tensor of type $(n, 0)$ while the input feature map $w$ is a tensor of type $(m, 0)$. The kernel $\psi$ is of type $(n, m)$. For example, one can use these types of $G$-convolutions to reduce a rank 2 tensor to a vector field within a G-CNN while keeping the equivariance property.

### 3.3 Split forms of tensor $G$-convolutions

As in the case of scalar $G$-convolutions, it is beneficial to write tensor $G$-convolutions Eqs. (48) in their split forms, analogous to Eqs. (32) and (38). Using $h = ys \in G$ with $y \in \mathbb{T}$ and $s \in G/\mathbb{T}$, we write the tensor field of type $(n, m)$

$$
w^I{}_J(h) = w^I{}_J(ys) = (w_s)^I{}_J(\mathbf{y}),
\tag{52}
$$

where $\mathbf{y} \in \mathbb{Z}^D$. Analogous to Eq. (35), left translations act on $w$ via

$$(L_g w)^I{}_J(h) = R^I{}_{I'}(w_{r^{-1}s})^{I'}{}_{J'}(R^{-1}(\mathbf{y} - \mathbf{x}))(R^{-1})^{J'}{}_J, \tag{53}$$

where $g = xr$, $R^I{}_J$ is the $(n, m)$ matrix representation of $r$ and $R$ is its representation on $\mathbb{Z}^D$. The two types of tensor $G$-convolutions can then be written as

$$\begin{aligned}
[\psi \star w]^I(g) &= \sum_{\mathbf{y} \in \mathbb{Z}^D} R^I{}_{I'} \psi^{I'}{}_{J'}(R^{-1}(\mathbf{y} - \mathbf{x}))(R^{-1})^{J'}{}_J w^J(\mathbf{y}) \\
&= [(L_r \psi)^I{}_J * w^J](\mathbf{x}),
\end{aligned} \tag{54}$$

$$\begin{aligned}
[\psi \star w]^I(g) &= \sum_{\mathbf{y} \in \mathbb{Z}^D} \sum_{s \in G/\mathbb{T}} R^I{}_{I'}(\psi_{r^{-1}s})^{I'}{}_{J'}(R^{-1}(\mathbf{y} - \mathbf{x}))(R^{-1})^{J'}{}_J w_s^J(\mathbf{y}) \\
&= \sum_{s \in G/\mathbb{T}} [(L_r \psi_{r^{-1}s})^I{}_J * (w_s)^J](\mathbf{x}),
\end{aligned} \tag{55}$$

where $\psi^I{}_J$ denotes the tensor components of the kernels, and $\psi^I{}_J : \mathbb{Z}^D \to \mathbb{R}$ for the first-layer and $\psi^I{}_J : G \to \mathbb{R}$ for the full $G$-Conv, respectively. Analogously, similar split forms can be obtained for tensors of mixed type and for $G$-convolutions that change the tensor type as in Eq. (51).

### 3.4 Implementing group equivariance in L-Convs for scalars and tensors

Having worked out $G$-convolutions for general tensors, we shift our focus back to gauge equivariance. The L-Conv introduced in Eq. (17) can alternatively be written as

$$[\psi * W](\mathbf{x}) = \sum_{\mathbf{y} \in \mathbb{Z}^D} \psi(\mathbf{y} - \mathbf{x}) U_{\mathbf{x} \to \mathbf{y}} W(\mathbf{y}) U_{\mathbf{y} \to \mathbf{x}}, \tag{56}$$

where we write $W(\mathbf{x})$ instead of $W_{\mathbf{x}}$ to denote a gauge dependent feature map. We keep the number of input and output channels set to one for convenience. The paths $\mathbf{x} \to \mathbf{y}$ in the subscripts of the links $U$ are kept general to further simplify notation. Since we only want to allow for straight paths, however, we set all other elements of the kernel $\psi$ to zero. Note that we only need to consider how the convolution acts on the local variables $W$ because the links are unaffected by an L-Conv.

The L-Conv operation is translationally equivariant, i.e. the gauge equivariant convolution commutes with translations. When acting on $[\psi * W]$ with a translation $z$, we find

$$\begin{aligned}
L_z[\psi * W](\mathbf{x}) = [\psi * W](\mathbf{x} - \mathbf{z}) &= \sum_{\mathbf{y} \in \mathbb{Z}^D} \psi(\mathbf{y} - \mathbf{x} + \mathbf{z}) U_{\mathbf{x} - \mathbf{z} \to \mathbf{y}} W(\mathbf{y}) U_{\mathbf{y} \to \mathbf{x} - \mathbf{z}} \\
&= \sum_{\mathbf{y}' \in \mathbb{Z}^D} \psi(\mathbf{y}' - \mathbf{x}) U_{\mathbf{x} - \mathbf{z} \to \mathbf{y}' - \mathbf{z}} W(\mathbf{y}' - \mathbf{z}) U_{\mathbf{y}' - \mathbf{z} \to \mathbf{x} - \mathbf{z}} \\
&= [\psi * L_z W](\mathbf{x}),
\end{aligned} \tag{57}$$

where the shift by $z$ induces a translation on both $W$ and $U$. We note that our notation for gauge equivariant convolutions explicitly hides the gauge links, but we want to stress that translations must also be applied to the links $U$ and that links should be viewed as part of the input. Keeping in mind the transformation properties of $U$ and $W$, given by Eqs. (9) and (15), it is straightforward to check that the L-Conv is equivariant under gauge transformations:

$$[\psi * T_\Omega W](\mathbf{x}) = \sum_{\mathbf{y} \in \mathbb{Z}^D} \psi(\mathbf{y} - \mathbf{x}) T_\Omega U_{\mathbf{x} \to \mathbf{y}} T_\Omega W(\mathbf{y}) T_\Omega U_{\mathbf{y} \to \mathbf{x}} = \Omega(\mathbf{x})[\psi * W](\mathbf{x})\Omega^\dagger(\mathbf{x}). \tag{58}$$

Thus, the L-Conv commutes both with translations on $\mathbb{Z}^D$ and with lattice gauge transformations.

Previously, we have determined the split forms of $G$-convolutions when the group $G$ is a semi-direct product of translations $\mathbb{T} \sim \mathbb{Z}^D$ and proper and improper rotations $G/\mathbb{T}$. For scalar feature maps, we found Eqs. (32)

and (38), which reduce the $G$-convolution to $\mathbb{Z}^D$-convolutions with rotated kernels. Following the same ideas, we now extend our lattice gauge equivariant convolutions so that they respect not only translations but larger symmetry groups $G$. If we consider $W : \mathbb{Z}^D \to \mathbb{C}^{N \times N}$ to transform as a scalar under $G$,

$$L_g W(\mathbf{x}) = W(g^{-1} \cdot \mathbf{x}) = W(R^{-1}(\mathbf{x} - \mathbf{y})), \tag{59}$$

then we can easily extend $G$-convolutions to gauge dependent fields $W$ by replacing the standard $\mathbb{Z}^D$-convolution with the L-Conv operation. For the first-layer $G$-Conv of Eq. (32) we then have

$$\begin{aligned}
[\psi \star W](xr) &= [L_r \psi * W](\mathbf{x}) \\
&= \sum_{\mathbf{y} \in \mathbb{Z}^D} \psi(R^{-1}(\mathbf{y} - \mathbf{x})) U_{\mathbf{x} \to \mathbf{y}} W(\mathbf{y}) U_{\mathbf{y} \to \mathbf{x}}.
\end{aligned} \tag{60}$$

The resulting feature map $\tilde{W}(g) = [\psi \star W](g)$ is now a feature map on the group $G$ and transforms under $G$ according to

$$L_g \tilde{W}(h) = \tilde{W}(g^{-1}h) = [\psi \star L_g W](h). \tag{61}$$

Additionally, under lattice gauge transformations we have

$$\begin{aligned}
T_\Omega \tilde{W}(g) &= T_\Omega \tilde{W}(xr) \\
&= [L_r \psi * T_\Omega W](xr) \\
&= \Omega(\mathbf{x})[L_r \psi * W](xr)\Omega^\dagger(\mathbf{x}) \\
&= \Omega(\mathbf{x})\tilde{W}(xr)\Omega^\dagger(\mathbf{x}),
\end{aligned} \tag{62}$$

i.e. the feature map $\tilde{W}(g) = \tilde{W}(xr)$ transforms locally at $\mathbf{x}$. Similarly, the full $G$-Conv is given by

$$[\psi \star W](g) = \sum_{h \in G} L_g \psi(h) U_{q(g) \to q(h)} W(h) U_{q(h) \to q(g)}, \tag{63}$$

where $q : G \to \mathbb{Z}^D$ projects group elements $g = xr$ down to the lattice by acting on the origin:

$$q(g) = g \cdot \mathbf{0} = x \cdot r \cdot \mathbf{0} = x \cdot \mathbf{0} = \mathbf{x}. \tag{64}$$

More compactly, both types of $G$-Conv can be written as

$$[\psi \star W](g) = \sum_{h \in H} L_g \psi(h) U_{q(g) \to q(h)} W(h) U_{q(h) \to q(g)}, \tag{65}$$

with $H = \mathbb{T} \sim \mathbb{Z}^D$ or $H = G$. Similarly, we can generalize these scalar $G$-convolutions to tensor convolutions in the same way as in the previous section. A general $G$-convolution for gauge dependent tensors is given by

$$\tilde{W}^I(g) = [\psi \star W]^I(g) = \sum_{h \in H} (L_g \psi(h))^I{}_J U_{q(g) \to q(h)} W^J(h) U_{q(h) \to q(g)}, \tag{66}$$

where $W$ and $\psi$ are of type $(n, 0)$ and $(n, m)$, respectively, and the output feature map transforms as a tensor of type $(m, 0)$. Note that gauge equivariant $G$-convolutions merely differ in the appearance of parallel transporters compared to Eq. (48). Consequently, the proof of equivariance of Eq. (66) under left translations in $G$ largely follows the steps performed in Eq. (50).

One aspect of $G$-equivariant L-Conv layers that has been missing in the discussion so far is the inclusion of bias terms. Generally, bias terms are additional terms added to the convolution,

$$\tilde{W}^I(g) = [\psi \star W]^I(g) + b^I(g), \tag{67}$$

where $b^I(g)$ is a tensor that is independent of the feature map $W^I(g)$. To retain group equivariance in Eq. (67), we require $b^I(g)$ to be invariant under left translations:

$$(L_k b)^I(g) = R^I{}_J b^J(k^{-1}g) \overset{!}{=} b^I(g), \qquad \forall k \in G. \tag{68}$$

Depending on the symmetry group $G$, the number of dimensions $D$, and the tensor type $(n,0)$ of the feature map, such invariant tensors may or may not exist. In the case of scalar feature maps, namely $W : \mathbb{Z}^D \to \mathbb{C}^{N \times N}$ (first layer) and $W : G \to \mathbb{C}^{N \times N}$ (deeper layers), bias terms simply correspond to unit matrices added to the output of the convolution:

$$\tilde{W}(g) = [\psi \star W](g) = \sum_{h \in H} (L_g \psi(h)) U_{q(g) \to q(h)} W(h) U_{q(h) \to q(g)} + b_0 \mathbb{1}, \tag{69}$$

where $b_0$ is a trainable parameter. However, the geometric structure of these terms becomes more complicated for general tensor feature maps and depends on the symmetry group $G$. For example, if $G$ consists of rotations, reflections, and translations, then there is no vector-type (i.e. tensor of type $(1,0)$) bias term. For rank 2 tensors, i.e. tensors of type $(2,0)$, bias terms correspond to Kronecker deltas $\delta_{ij}$. In $D = 3$ dimensions, rank 3 bias terms may be given by the Levi-Civita tensor $\epsilon_{ijk}$ assuming that reflections are not considered as symmetries. More generally for the rotation group in $D = 3$, higher ranks are given by products and contractions of $\delta_{ij}$ and $\epsilon_{ijk}$.

### 3.5 $G$-equivariant bilinear layers

Bilinear layers, which map two feature maps into one, can be generalized to respect $G$-equivariance. Without including channels, a $G$-equivariant bilinear layer for tensors reads

$$W^I(g) = (L_g \hat{\psi})^I{}_{JK} V_1^J(g) V_2^K(g), \tag{70}$$

where $V_1$ and $V_2$ are tensor feature maps of type $(n_1, 0)$ and $(n_2, 0)$, respectively. The product of $V_1$ and $V_2$ is understood to be a matrix product with respect to their $\mathbb{C}^{N \times N}$ matrix structure. The weight tensor $\hat{\psi}$ is of type $(m, n_1 + n_2)$ and the resulting tensor feature map $W$ is of type $(m, 0)$. Note that the weight tensor is constant, i.e. it does not depend on the group element $g$. However, because of its index structure, it transforms in the usual way

$$(L_g \hat{\psi})^I{}_{JK} = R^I{}_{I'} \hat{\psi}^{I'}{}_{J'K'} (R^{-1})^{J'}{}_J (R^{-1})^{K'}{}_K, \tag{71}$$

where $R$ is the matrix representation of the rotational part $r$ of $g = xr$. It follows that the bilinear layer is equivariant under left translations:

$$(L_k W)^I(g) = (L_g \hat{\psi})^I{}_{JK} (L_k V_1)^J(g) (L_k V_2)^K(g). \tag{72}$$

Gauge equivariance follows from the fact that when both $V_1$ and $V_2$ are locally transforming as in Eq. (11), their matrix product also transforms locally. Consequently, the resulting feature map $W^I$ transforms locally as well.

The bilinear layer can be expressed as a special case of a gauge equivariant tensor convolution. Consider a tensor $V$ of type $(n_1 + n_2, 0)$ that factorizes into two tensors $V_1$ and $V_2$ with types $(n_1, 0)$ and $(n_2, 0)$:

$$V^{JK} = V_1^J V_2^K. \tag{73}$$

A general lattice gauge equivariant convolution applied to $V$ then reads

$$W^I(g) = \sum_{h \in H} (L_g \psi)^I{}_{JK}(h) U_{q(g) \to q(h)} V_1^J(h) V_2^K(h) U_{q(h) \to q(g)}. \tag{74}$$

The bilinear layer is local, i.e. feature maps are all evaluated at the same element $g$. We therefore assume that the kernel has the particular form

$$\psi^I{}_{JK}(h) = \hat{\psi}^I{}_{JK} \delta(h), \tag{75}$$

where $\delta(e) = 1$ and $\delta(h) = 0$ for $h \neq e$ with the unit element $e$, and $\hat{\psi}$ is a constant tensor. The left translated kernel $L_g \psi(h)$ only has a single non-zero contribution to the sum, namely when $g^{-1}h$ is the unit element $e$ of the group, i.e. $h = g$. The convolution then reduces to Eq. (70), because the parallel transporters reduce to unit matrices for constant paths $q(g) \to q(g)$:

$$U_{q(g) \to q(g)} = \mathbb{1}. \tag{76}$$

Bias terms may also be included in the bilinear layer. As in the case of $G$-equivariant L-Conv layers, bias terms must be invariant tensors.

### 3.6 Trace layers

It is straightforward to show that trace layers are compatible with $G$-equivariance. Given a locally gauge transforming tensor field $W^I(g)$ we define the traced tensor $w^I(g)$ simply as

$$w^I(g) = \text{Tr}[W^I(g)], \tag{77}$$

where the trace is taken over the $\mathbb{C}^{N \times N}$ matrix structure. The trace yields a gauge invariant tensor, which can be shown via

$$
\begin{aligned}
T_\Omega w^I(g) &= \text{Tr}[T_\Omega W^I(g)] \\
&= \text{Tr}[\Omega(\mathbf{x})W^I(g)\Omega^\dagger(\mathbf{x})] \\
&= \text{Tr}[W^I(g)] \\
&= w^I(g),
\end{aligned}
\tag{78}
$$

where we have assumed $q(g) = \mathbf{x}$. The traced tensor transforms under $G$ as a tensor because the left translation $L_k, \forall k \in G$ commutes with the trace operation over $\mathbb{C}^{N \times N}$. Note that the trace layer does not have any trainable parameters. It can be used at the end of an L-CNN to obtain gauge invariant scalars and tensors, which can then be further processed by standard group equivariant networks.

### 3.7 Activation functions

The purpose of an activation function is to introduce non-linearity into the network. In a CNN or G-CNN with scalar input, it is usually applied point-wise, i.e. $f'(\mathbf{x}) = \nu(f(\mathbf{x}))$, where $f, f' : G \to \mathbb{R}$ are the feature maps before and after applying the (scalar) activation function $\nu : \mathbb{R} \to \mathbb{R}$, respectively. When the input feature map is a general tensor field $w$ of type $(n, 0)$, we choose an ansatz similar to Eq. (23), namely

$$w'^I = C_\nu w^I = \nu(w)w^I, \tag{79}$$

with the operator $C_\nu$, which applies the activation function to the tensor. For $G$-equivariance to hold, $C_\nu$ has to commute with $L_g$

$$C_\nu L_g w^I = L_g C_\nu w^I. \tag{80}$$

Keeping in mind the transformation property of $w$, which is given by Eq. (49), we find

$$\nu(\widetilde{w}(h)) = \nu(w(h)), \tag{81}$$

where $\widetilde{w}^I(h) = R^I{}_{I'}w^{I'}(h)$. Thus, the activation function $\nu$ has to be invariant under transformations in the group $G$.

The generalization of Eq. (79) to lattice gauge invariant activation functions is straightforward. We make the ansatz

$$W'^I(g) = \nu(w(g))W^I(g), \tag{82}$$

with the local variables $W^I$ and $W'^I$ before and after the application of the activation function, respectively, and $w^I(g) = \text{Re}(\text{Tr}(W^I(g)))$. The form of $w(g)$ guarantees equivariance under lattice gauge transformations, and Eq. (81) guarantees equivariance under transformations in $G$.

A possible choice for an activation function is a norm non-linearity (Gerken et al., 2023)

$$\nu(w(g)) = \alpha(\|w(g)\|), \tag{83}$$

with a norm that satisfies Eq. (81), such as $\|w(g)\| = \sqrt{\sum_I (w^I(g))^2}$. Since the output of a norm is always non-negative, choosing the Heaviside step function $\Theta$ as the function $\alpha$, which we have done in Section 2 to mimic the well-known ReLU activation function, would not lead to a non-linearity. Therefore, we introduce a trainable bias $b \geq 0$ and set

$$\alpha(\|w(g)\|) = \Theta(\|w(g)\| - b). \tag{84}$$

In the above example, the ReLU activation function becomes active if the norm of $w$ exceeds the bias $b$.

### 3.8 Pooling

Pooling layers, which are often used to reduce the domain of a feature map, can also be generalized to G-CNNs. In analogy to Cohen & Welling (2016), we split pooling layers into separate pooling and subsampling steps. The pooling step performs a convolution-like operation on the feature map, but does not change the domain. The domain reduction happens during the subsequent subsampling step with a particular stride. Typically, subsampling leads to a reduction in symmetry, depending on the stride. We first review this procedure for scalar feature maps on the group $G$ before generalizing it to tensor fields on $G$ and finally to the L-CNN.

As a motivating example, we consider sum pooling. In a traditional CNN in two dimensions, sum pooling is performed by summing up all values of a feature map in a pooling domain $\mathcal{D} \subset \mathbb{Z}^2$, e.g. a $2 \times 2$ region, which is moved across the lattice. It differs from average pooling only by a constant factor determined by the cardinality of $\mathcal{D}$. Since sum pooling in a traditional CNN can be written as a special case of convolution (every kernel coefficient in the pooling region is set to one), it is equivariant with respect to translations. Analogously, sum pooling on a feature map $f : G \to \mathbb{R}$ can be viewed as a special case of the full $G$-convolution in Eq. (33), when setting the kernel $\psi(g)$ to one in the pooling domain $\mathcal{D} \subset G$ and zero elsewhere:

$$f'(g) = \sum_{h \in G} \psi(g^{-1}h)f(h) = \sum_{h \in g\mathcal{D}} f(h). \tag{85}$$

Here, $g\mathcal{D} = \{gd : d \in \mathcal{D}\}$ refers to the $g$-translated pooling region and $f' : G \to \mathbb{R}$ is the feature map after the pooling step. Another well-known pooling operation is max pooling, given by

$$f'(g) = \max_{h \in g\mathcal{D}} f(h). \tag{86}$$

Sum and max pooling can be generalized to other pooling operations by introducing an operator $P$ that acts on a feature map $f$ by

$$f'(g) = (Pf)(g) = \mathcal{P}(f(gd_1), f(gd_2), \ldots, f(gd_N)), \tag{87}$$

where $\mathcal{P} : \mathbb{R}^N \to \mathbb{R}$ is a function, and $N$ is the cardinality of $\mathcal{D} = \{d_1, d_2, \ldots, d_N\}$. The operator $P$ $g$-translates the set $\mathcal{D}$ over the feature map, so the same pooling operation is performed everywhere on $G$, rendering it $G$-equivariant. That is, it commutes with the left translation operator $L_k$. This can be explicitly shown via

$$
\begin{aligned}
L_k(Pf)(g) &= (Pf)(k^{-1}g) \\
&= \mathcal{P}(f(k^{-1}gd_1), \ldots, f(k^{-1}gd_N)) \\
&= \mathcal{P}(L_k f(gd_1), \ldots, L_k f(gd_N)) \\
&= (PL_k f)(g).
\end{aligned}
\tag{88}
$$

To achieve a pooling with stride $s$ in a traditional CNN, after the pooling step, the feature map is subsampled on the subgroup $\mathbb{T}_s \subset \mathbb{T}$ consisting of all translations that are multiples of $s$ elementary translations. The resulting feature map is then equivariant under $\mathbb{T}_s$. Analogously, for feature maps $f : G \to \mathbb{R}$, subsampling is performed on a subgroup $H \subset G$. This procedure is known as subgroup pooling and the resulting feature map retains equivariance only under the subgroup $H$.

If the pooling region $\mathcal{D}$ is itself a subgroup of $G$, then $g\mathcal{D}$ are left cosets of $\mathcal{D}$. They partition the group $G$ into disjoint, equally sized subsets. Since the left cosets are invariant under the right-action (or right translation) of $\mathcal{D}$, i.e. $gd\mathcal{D} = g\mathcal{D}$, $\forall d \in \mathcal{D}$, the corresponding parts of the feature map are invariant under said action as well, and we can pick one such part to be the resulting feature map without losing $G$-equivariance. This type of pooling is called coset pooling. For example, in a network with scalar input on $\mathbb{Z}^D$ that is promoted to the group $G = \mathbb{T} \rtimes K$ by a first-layer $G$-convolution and further convolved with full $G$-convolutions, coset pooling over $K$ would yield a $G$-equivariant feature map on $\mathbb{T} \sim \mathbb{Z}^D$.

In order to generalize sum pooling to tensor fields $w(g)$ on $G$, we take full tensor $G$-convolutions, which are given by Eq. (48), with $H = G$ as a starting point and set the kernel $\psi^I{}_J(g) = \delta^I{}_J \psi(g)$, where $\psi(g)$ is one

if $g \in \mathcal{D}$ and zero everywhere else. We get

$$
\begin{aligned}
w'^I(g) = [\psi \star w]^I(g) &= \sum_{h \in G} (L_g \psi)^I{}_J(h) w^J(h) \\
&= \sum_{h \in G} R^I{}_{I'} \psi^{I'}{}_{J'}(g^{-1}h)(R^{-1})^{J'}{}_J w^J(h) \\
&= \sum_{h \in G} R^I{}_{I'} \delta^{I'}{}_{J'} \psi (R^{-1})^{J'}{}_J w^J(h) \\
&= \sum_{h \in G} \psi \, w^I(h) = \sum_{h \in g\mathcal{D}} w^I(h),
\end{aligned}
\tag{89}
$$

which differs from sum pooling in the scalar case, Eq. (85), only by the tensor index $I$. We take this resemblance as motivation to define the general pooling operator $P$ on tensor fields of type $(n, 0)$, i.e. $w : G \to V$ with $V = (\mathbb{R}^D)^n$, as

$$
w'^I(g) = (Pw)^I(g) = \mathcal{P}(w(gd_1), \dots, w(gd_N))^I,
\tag{90}
$$

where the function $\mathcal{P}$ maps $\mathcal{P} : V^N \to V$. The pooling operation commutes with left translations if

$$
\mathcal{P}(Rw(d_1), \dots, Rw(d_N))^I = R^I{}_{I'} \mathcal{P}(w(d_1), \dots, w(d_N))^{I'},
\tag{91}
$$

which means that if the pooling operation commutes with the outer transformation $R$, then it is $G$-equivariant. This is the case for sum pooling defined in Eq. (89). Furthermore, if the norm $\|\cdot\|$ is unaffected by the outer transformation $R$, then $G$-equivariance also holds for the max pooling operation

$$
w'^I(g) = \max_{h \in g\mathcal{D}} w^I(h) = w^I(g'),
\tag{92}
$$

where

$$
g' = \operatorname*{argmax}_{h \in g\mathcal{D}} \|w(h)\|,
\tag{93}
$$

corresponds to the element $g'$ that maximizes the norm of $w$ in the translated pooling domain $g\mathcal{D}$. [1] The above definitions for tensor feature maps $w : G \to V$ are consistent with the corresponding definitions for scalar feature maps $f : G \to \mathbb{R}$, where $V = \mathbb{R}$ and the rotation matrices $R$ reduce to the identity operation.

To generalize sum pooling of tensor feature maps to L-CNNs, we start with the L-CNN full tensor $G$-convolution, which is given by Eq. (66), with $H = G$. We set the kernel $\psi^I{}_J(g) = \delta^I{}_J \psi(g)$, where $\psi(g)$ is one inside the pooling domain and zero elsewhere. Steps analogous to Eq. (89) lead to

$$
W'^I(g) = \sum_{h \in g\mathcal{D}} U_{q(g) \to q(h)} W^I(h) U_{q(h) \to q(g)} = \sum_{h \in g\mathcal{D}} W_g^I(h),
\tag{94}
$$

where $W$ and $W'$ are the local variables before and after the pooling step, respectively, and

$$
W_g^I(h) = U_{q(g) \to q(h)} W^I(h) U_{q(h) \to q(g)},
\tag{95}
$$

denotes the variable $W$, parallel transported from $h$ to $g$. Note that the fact that we only consider straight paths in the convolution restricts the shape of the pooling region $\mathcal{D}$. In principle, however, it can be chosen arbitrarily as long as the paths $q(h) \to q(g)$ of the parallel transporters $U_{q(h) \to q(g)}$ are chosen accordingly. A general pooling step, represented by the aforementioned pooling operator $P$, on local tensor variables $W$ can be written as

$$
W'^I(g) = (PW)^I(g) = \mathcal{P}(W_g(gd_1), \dots, W_g(gd_N))^I,
\tag{96}
$$

where $\mathcal{P} : V^N \to V$. It is $G$-equivariant if the outer transformation commutes with the pooling operation

$$
\mathcal{P}(RW_g(d_1), \dots, RW_g(d_N))^I = R^I{}_{I'} \mathcal{P}(W_g(d_1), \dots, W_g(d_N))^{I'}.
\tag{97}
$$

---

[1] For simplicity, we assume that the maximum of the feature map is unique. In practical applications of G-CNNs and L-CNNs, it is highly unlikely to encounter cases where the maximum cannot be uniquely determined.

Similarly, we require the pooling layer to be equivariant under lattice gauge transformations

$$T_\Omega(PW)^I(g) = \Omega(\mathbf{x})(PW)^I(g)\Omega^\dagger(\mathbf{x}) \stackrel{!}{=} (PT_\Omega W)^I(g), \tag{98}$$

which implies that the function $\mathcal{P}$ must also satisfy

$$\mathcal{P}\big(\Omega W_g(d_1)\Omega^\dagger, \dots, \Omega W_g(d_N)\Omega^\dagger\big)^I = \Omega\,\mathcal{P}\big(W_g(d_1), \dots, W_g(d_N)\big)^I\,\Omega^\dagger. \tag{99}$$

We omitted the argument $\mathbf{x} = q(g) = q(xr)$ of $\Omega = \Omega(\mathbf{x})$ for simplicity. For example, max pooling can be defined as

$$W'^I(g) = \max_{h \in g\mathcal{D}} W_g^I(h) = W_g^I(g'), \tag{100}$$

with

$$g' = \operatorname*{argmax}_{h \in gD} \big\| \operatorname{Re}\big(\operatorname{Tr}\big(W^I(h)\big)\big)\big\|, \tag{101}$$

where the trace leads to the parallel transporters dropping out. Clearly, this form of max pooling satisfies both $G$-equivariance and equivariance under gauge transformations.

As a concrete example of how these pooling layers can be used, consider a network that uses local tensor variables $W^I$ on $\mathbb{Z}^D$ as input. In the first layer, the input feature maps are promoted to the group $G = \mathbb{T} \rtimes K$ via a first-layer lattice $G$-convolution. After that, the feature maps are convolved by full lattice $G$-convolutions. Coset pooling over $K$ can then be applied to reduce the domain of the feature maps from $G$ back to $\mathbb{Z}^D$ while retaining both gauge and global group symmetry. For example, if we use sum pooling, each resulting feature map reads

$$W'^I(g) = \sum_{h \in gK} W^I(h), \tag{102}$$

where the parallel transporters are trivial because the projections $q(g) = q(h) = \mathbf{x}$ coincide at the same point $\mathbf{x} \in \mathbb{Z}^D$. The feature map is invariant in the sense of $W'^I(xr) = W'^I(xr')$ for $r, r' \in K$. Subsampling in $K$ thus leads to a $G$-equivariant tensor $W'^I(\mathbf{x})$ on $\mathbb{Z}^D$. In this way, $G$-equivariant L-CNNs can be used to model gauge and group equivariant functions which map input feature maps on the lattice $\mathbb{Z}^D$ to new output feature maps on $\mathbb{Z}^D$.

### 3.9 Computational requirements of $G$-equivariant L-CNNs

Having defined the $G$-equivariant generalizations of relevant L-CNN layers, we need to comment on the computational resources required by these layers. There are two main differences to the original formulation of L-CNNs: the first concerns the domain on which feature maps are defined, and the second concerns the computational complexity of a $G$-convolution layer.

Regarding the first point, we recall that in standard L-CNNs, feature maps are functions on the lattice $\mathbb{Z}^D$ (with periodic boundary conditions) consisting of $N_{\mathrm{ch}} \cdot N_l^D$ complex matrices in each layer, where $N_{\mathrm{ch}}$ is the number of channels and $N_l^D$ is the total number of lattice sites on the hypercubic lattice. Depending on the dimensions of the lattice, our $G$-equivariant generalizations enlarge these domains considerably. After the first-layer $G$-convolution, feature maps are promoted to functions on the global symmetry group $G$. Since $G$ is a semi-direct product of the translation group $\mathbb{T} \sim \mathbb{Z}^D$ and the stabilizer group $K$, we may write the number of group elements as $|G| = |K| N_l^D$. Feature maps on $G$ may thus be represented by $N_{\mathrm{ch}} \cdot |K| \cdot N_l^D$ complex matrices, which leads to an increased memory requirement by a factor of $|K|$ compared to the original L-CNN formulation. For example, if we consider rotoreflections in $D = 2$, we have $|K| = 4 \cdot 2 = 8$ group elements since there are four possible rotations and two mirror operations about the two axes. For $D = 3$ and $D = 4$ we have $|K| = 48$ (octahedral group) and $|K| = 384$ (hyperoctahedral group), respectively. These factors are large, considering the fact that the models used in the original L-CNN study (Favoni et al., 2022) easily saturated the available memory on modern GPUs. For the physically relevant case of $D = 4$ it may be argued that the symmetry subgroup $K$ may be smaller than the full hyperoctahedral group. In practice, one typically uses lattice sizes $N_t \cdot N_l^3$, where $N_t$ is the number of cells along the time directions with $N_t \neq N_l$, which reduces the symmetry. Thus, the group $K$ should consist of rotations and

reflections in the spatial directions (octahedral group) and reflections along the fourth axis only, which leads to $|K| = 2 \cdot 48 = 96$. Nevertheless, the memory footprint of $G$-equivariant L-CNNs is generically much larger than of their $\mathbb{Z}^D$-equivariant counterparts.

Additionally, one has to consider the computational complexity of a $G$-convolution layer, i.e. the number of operations required to evaluate such convolutions. Similar to traditional CNNs, the original L-Conv layer consists of a sum over the lattice $\mathbb{Z}^D$. We typically restrict the kernel of the L-Conv to the lattice axes, such that in total $D \cdot (N_k - 1) + 1$ terms have to be summed over to compute the result of the convolution. $G$-convolutions extend this sum to run over the full symmetry group $G$. Thus, the number of terms to consider is larger by a factor of $|K|$.

### 3.10 Universality of lattice gauge equivariant layers

Neural network architectures such as the multilayer perceptron (MLP) (Hornik et al., 1989), equivariant MLPs (Ravanbakhsh, 2020), CNNs (Zhou, 2020) and G-CNNs (Sonoda et al., 2022) exhibit universality: given enough depth (number of layers) or enough width (number of channels in a layer), these networks can approximate any function in a suitable function space to arbitrary precision. One may ask to what extent these universality properties also apply to lattice gauge equivariant networks.

Favoni et al. (2022) show that an L-CNN consisting of alternating convolutional and bilinear layers can be used to construct any Wilson loop, that is, any closed path of gauge links. Combining this construction of arbitrary loops with non-linear activation layers, a trace layer, and a CNN as a final network, the L-CNN can represent any non-linear combination of loops. With the result of Durhuus (1980), which shows that the linear span of products of traced Wilson loops is dense in the space of continuous gauge invariant functions, one can conclude that the L-CNN is a universal approximator for continuous gauge invariant functions on the lattice. Ostensibly, one can perform the same explicit construction using $G$-equivariant layers, which are generalizations of the original $\mathbb{Z}^D$-equivariant L-CNN layers.

A related question is the universality of individual layers. For example, is the $G$-equivariant L-Conv layer the most general $G$-equivariant, gauge equivariant *linear* map with respect to the feature map $W$? It appears that this is not the case due to the path dependence of parallel transporters employed in the convolution. The non-universality of the linear layers can be demonstrated for the simpler case $G = \mathbb{Z}^D$. In this work, we have always assumed that the paths along which feature maps are transported must be straight lines. Clearly, in order to describe arbitrary linear maps, one must consider a generalization of the usual $\mathbb{Z}^D$-equivariant L-Conv (see Eq. (56), which we repeat here for convenience)

$$[\psi * W](\mathbf{x}) = \sum_{\mathbf{y} \in \mathbb{Z}^D} \psi(\mathbf{y} - \mathbf{x}) U_{\mathbf{x} \to \mathbf{y}} W(\mathbf{y}) U_{\mathbf{y} \to \mathbf{x}}, \tag{103}$$

with the usual restriction of $\psi$ to only allow straight paths straight paths $\mathbf{x} \to \mathbf{y}$. In principle, arbitrary paths from $\mathbf{x}$ to $\mathbf{y}$ are allowed, which leads to

$$[\psi * W]'(\mathbf{x}) = \sum_{\mathbf{y} \in \mathbb{Z}^D} \sum_{c \in \mathcal{C}_{\mathbf{y} \to \mathbf{x}}} \psi_c U_c W(\mathbf{y}) U_c^{-1}, \tag{104}$$

where the sum runs over the set of paths $\mathcal{C}_{\mathbf{y} \to \mathbf{x}}$ which start at $\mathbf{y}$ and end at $\mathbf{x}$. The kernel $\psi$ must be promoted to a map $\psi : \mathcal{C} \to \mathbb{R}$, where $\mathcal{C}$ is the set of all paths on the lattice. We note that translation equivariance requires that the kernel must itself be invariant under shifts, i.e. if two paths $c$ and $c'$ are equivalent up to a translation, then it holds that $\psi_c = \psi_{c'}$.

There is an obvious practical problem with Eq. (104): the sum over paths requires (countably) infinite terms for full generality. Even if one restricts the kernel to a finite receptive field, i.e. allowing only paths that stay close to the base point $\mathbf{x}$, there are still infinitely many multi-winding loops to account for, and thus infinitely many kernel coefficients. This is surprising since a standard $\mathbb{Z}^D$-convolution can be represented by a finite number of trainable parameters when restricted to a finite receptive field. Clearly, linear gauge equivariant layers such as the $\mathbb{Z}^D$-equivariant L-Conv are more complicated because they are only linear with respect to feature maps $W$, but are generally non-linear functions of link variables $U$, which make up the

parallel transporters appearing in the convolution. For practical reasons, it is sensible to restrict the set of paths, but this necessarily limits the expressivity of individual linear layers and renders them non-universal. We note that this ambiguity in the choice of paths is not unique to the L-CNN. Gauge equivariant layers on smooth manifolds (see e.g. Cohen (2021); Weiler et al. (2023)) typically use geodesics for parallel transport, but any other choice would be compatible with gauge equivariance. Within lattice gauge theory, a practical example along the lines of Eq. (104) can be found in Lehner & Wettig (2023a), where convolutional layers over a restricted set of paths are applied to a regression problem.

## 4 L-CNNs from a bundle theoretic viewpoint

Having laid out the details of how to extend L-CNNs to larger global symmetries in previous sections, we now focus on the mathematical foundations of gauge equivariant convolutional neural networks and how they relate to the translationally equivariant L-CNN. There is a mathematical theory of equivariant neural networks that uses fiber bundles to describe symmetries and to capture geometric information in data (Cohen et al., 2019a; Aronsson, 2022). This theoretical framework models data points as fields or, more generally, as sections of vector bundles associated to a principal bundle that specifies relevant symmetries. In Sections 4.1-4.3 we show that the original L-CNN (Favoni et al., 2022) is a discretization of a continuous model within this theory. Section 4.4 looks at how the original L-CNN can be directly generalized to other representations, i.e. beyond locally transforming matrices $W(\mathbf{x})$ as input. Finally, in Section 4.5, we discuss the possibility of placing fully group-equivariant L-CNNs into this theoretical framework.

### 4.1 Bundle formalism

Geometric deep learning, and equivariant neural networks in particular, uses fiber bundles because they allow nontrivial global geometries; fiber bundles generalize the (geometrically trivial) product $\mathcal{M} \times F$ between two spaces $\mathcal{M}$ and $F$, respectively known as the base space and the characteristic fiber. We visualize this product as attaching a fiber $\{\mathbf{x}\} \times F$ to each point $\mathbf{x} \in \mathcal{M}$ of the base space. A general fiber bundle is a collection of fibers

$$E = \bigcup_{\mathbf{x} \in \mathcal{M}} F_{\mathbf{x}}, \tag{105}$$

where each fiber $F_{\mathbf{x}} \simeq F$ in the total space $E$ is equivalent to the characteristic fiber. There is also a projection $\pi : E \to \mathcal{M}$ that maps each element $p \in F_{\mathbf{x}} \subset E$ to the point $\pi(p) = \mathbf{x}$ where its fiber is attached. Although the total space $E$ can be a more complicated object than a trivial bundle $\mathcal{M} \times F$, it is required to look like a product $\mathcal{D} \times F$ on certain local regions $\mathcal{D} \subset \mathcal{M}$. A common example is the comparison between a Möbius strip and a cylinder $S^1 \times [0, 1]$, where $S^1$ is the circle. Locally, both of these objects look like segments $\mathcal{D} \times [0, 1]$ for $\mathcal{D} \subset S^1$ but they have different global geometry. The Möbius strip is thus a nontrivial fiber bundle.

Principal bundles are fiber bundles such that the characteristic fiber is a group $K$, often called the structure group. Since the total space consists of fibers $K_{\mathbf{x}} \simeq K$, the structure group acts as an internal degree of freedom at each point $\mathbf{x} \in \mathcal{M}$ and principal bundles are therefore used to study local gauge symmetry.

Consider the trivial principal bundle on $\mathcal{M} = \mathbb{R}^D$ with structure group $K = \mathrm{SU}(N)$,

$$\pi : \mathbb{R}^D \times \mathrm{SU}(N) \to \mathbb{R}^D, \qquad \pi(\mathbf{x}, \Omega) = \mathbf{x}, \tag{106}$$

which we often refer to as $P = \mathbb{R}^D \times \mathrm{SU}(N)$. This bundle describes an $SU(N)$ gauge symmetry that is incorporated into fields over $\mathbb{R}^D$ in the following way: Let $V$ be a linear space whose elements transform according to a linear representation $\rho$ of $SU(N)$,

$$v \mapsto \rho(\Omega)v. \tag{107}$$

Triples $(\mathbf{x}, \Omega, v)$ are interpreted as an element $v \in V$ located at the position $\mathbf{x} \in \mathbb{R}^D$ and expressed in the gauge $\Omega \in \mathrm{SU}(N)$. The transformation in Eq. (107) means that if we let the identity matrix $\mathbb{1} \in SU(N)$ represent an initial gauge, then a gauge transformation $\mathbb{1} \mapsto \Omega$ can be achieved by transforming $v \mapsto \rho(\Omega)v$

instead, hence the triples $(\mathbf{x}, \Omega, v)$ and $(\mathbf{x}, \mathbb{1}, \rho(\Omega)v)$ are gauge equivalent. We can remove the gauge degree of freedom by defining an equivalence relation

$$(\mathbf{x}, \Omega, v) \sim (\mathbf{x}, \mathbb{1}, \rho(\Omega)v) \tag{108}$$

and considering the set of equivalence classes $E_\rho = P \times_\rho V = (P \times V)/\sim$. The equivalence class

$$[\mathbf{x}, \Omega, v] = [\mathbf{x}, \mathbb{1}, \rho(\Omega)v] \tag{109}$$

is thus a gauge invariant expression for the element $v \in V$ at the position $\mathbf{x} \in \mathbb{R}^D$.

By construction, the quotient space $E_\rho$ is a so-called associated bundle

$$\pi_\rho : E_\rho \to \mathbb{R}^D, \qquad \pi_\rho([\mathbf{x}, \Omega, v]) = \mathbf{x}, \tag{110}$$

and such bundles are used extensively in the mathematical theory of equivariant neural networks. For example, the inputs to and outputs from an equivariant neural network are called data points and are defined as sections of associated bundles, i.e. generalized fields $s : \mathbb{R}^D \to E_\rho$ of the form

$$s(\mathbf{x}) = [\mathbf{x}, \Omega, v(\mathbf{x})] = [\mathbf{x}, \mathbb{1}, \rho(\Omega)v(\mathbf{x})], \tag{111}$$

where $\Omega$ runs over all values of $\Omega \in SU(N)$ by definition of the equivalence class. The idea behind this definition is that instead of using feature maps (which depend on the choice of gauge), we use gauge-invariant data points $s$ as input data, which map from the base space into the class of equivalent triplets. Thus, one can consider $s(\mathbf{x})$ at some point $\mathbf{x}$ as a gauge invariant object and use data points to describe vector and tensor fields in a geometric (gauge and coordinate independent) way. In contrast, a feature map of locally transforming matrices $W(\mathbf{x})$ is a single representative of the gauge-invariant data point $s_W(\mathbf{x})$. Similarly, one may also consider gauge-dependent tensor fields $W^I(\mathbf{x})$ as representatives of their respective data points.

Each fiber $E_\mathbf{x}$ of the associated bundle $E_\rho$ is a linear space with respect to linear combinations

$$\alpha[\mathbf{x}, \Omega, v] + \alpha'[\mathbf{x}, \Omega, v'] = [\mathbf{x}, \Omega, \alpha v + \alpha' v'], \tag{112}$$

for scalars $\alpha, \alpha'$ and $v, v' \in V$. We can therefore take pointwise linear combinations $\alpha s(\mathbf{x}) + \alpha' s'(\mathbf{x})$ of data points, making the set $\Gamma(\rho)$ of all data points $s : \mathbb{R}^D \to E_\rho$ into a linear space.

Layers in an equivariant neural network are maps

$$\Phi : \Gamma(\rho_1) \to \Gamma(\rho_2), \tag{113}$$

between the spaces of data points for two possibly different representations $(\rho_1, V_1)$ and $(\rho_2, V_2)$ of the structure group $SU(N)$. Layers can be either linear or nonlinear, and we say that $\Phi$ is gauge equivariant if it commutes with gauge transformations

$$\begin{aligned} T_\Omega s(\mathbf{x}) &= T_\Omega[\mathbf{x}, \tilde{\Omega}, v(\mathbf{x})] \\ &= [\mathbf{x}, \tilde{\Omega}\Omega(\mathbf{x}), v(\mathbf{x})] \\ &= [\mathbf{x}, \tilde{\Omega}, \rho(\Omega^\dagger(\mathbf{x}))v(\mathbf{x})]. \end{aligned} \tag{114}$$

We define gauge equivariant neural networks as compositions of gauge equivariant layers such as Eq. (113). Note that this definition makes no mention of convolutional layers or translation equivariance, allowing it to be used even in the absence of global symmetry. Convolutional layers are one of possibly many different types of layers. Moreover, (non-linear) activation functions, or the composition of a linear transformation and an activation function, are also considered layers under this definition as there is no requirement of linearity in Eq. (113). In the following, we will investigate how the original L-CNN relates to this theory.

## 4.2 Locally transforming variables

We claim that the locally transforming variables used in the original L-CNN are directly related to data points of the associated bundle $E_{\text{Ad}} = P \times_{\text{Ad}} \mathbb{C}^{N \times N}$. Here, $V = \mathbb{C}^{N \times N}$ is the linear space of complex $N \times N$-matrices and $\rho = \text{Ad}$ is the adjoint representation

$$\text{Ad}(\Omega) : W \mapsto \Omega W \Omega^\dagger, \qquad W \in \mathbb{C}^{N \times N}. \tag{115}$$

In order to investigate how the data points, given by Eq. (111), are related to $W(\mathbf{x})$ for this bundle, we fix a (local) gauge

$$\omega : \mathcal{D} \to P, \qquad \mathcal{D} \subseteq \mathbb{R}^D, \tag{116}$$

i.e. a local section of the principal bundle $P = \mathbb{R}^D \times SU(N)$. This principal bundle is trivial, and thus the gauge is given by $\omega(\mathbf{x}) = (\mathbf{x}, g(\mathbf{x}))$ for a unique function $g : \mathcal{D} \to SU(N)$. If $s : \mathbb{R}^D \to E_{\text{Ad}}$ is a data point, then the gauge selects a specific representative

$$f(\mathbf{x}) = (\mathbf{x}, g(\mathbf{x}), W(\mathbf{x})) \in P \times \mathbb{C}^{N \times N} \tag{117}$$

of the equivalence class $s(\mathbf{x})$ for $\mathbf{x} \in \mathcal{D}$. The triple in Eq. (117) describes a matrix $W(\mathbf{x}) \in \mathbb{C}^{N \times N}$, placed at the position $\mathbf{x} \in \mathbb{R}^D$ and expressed in the gauge $g(\mathbf{x})$. As our notation indicates, we argue that $W(\mathbf{x})$ is the matrix-valued, locally transforming variable used in the original L-CNN. For the purpose of verifying the transformation behavior of $W(\mathbf{x})$ under gauge transformations, we fix a second gauge $\omega' : \mathcal{D}' \to P$, given by $\omega'(\mathbf{x}) = (\mathbf{x}, g'(\mathbf{x}))$, and use it to select a representative

$$f'(\mathbf{x}) = (\mathbf{x}, g'(\mathbf{x}), W'(\mathbf{x})) \tag{118}$$

of the equivalence class $s(\mathbf{x}) \in E_\mathbf{x}$ for $\mathbf{x} \in \mathcal{D}'$. For each $\mathbf{x}$ in the intersection $\mathcal{D} \cap \mathcal{D}'$, Eqs. (117) and (118) select possibly different representatives $f(\mathbf{x}) \sim f'(\mathbf{x})$ of the same equivalence class $s(\mathbf{x})$ and must therefore be related by

$$(\mathbf{x}, g'(\mathbf{x}), W'(\mathbf{x})) = (\mathbf{x}, g(\mathbf{x})\Omega^\dagger(x), \Omega(x)W(\mathbf{x})\Omega^\dagger(\mathbf{x})). \tag{119}$$

Here, $\Omega(\mathbf{x}) = g'^\dagger(\mathbf{x})g(\mathbf{x})$ is the gauge transformation that transforms between $\omega$ and $\omega'$. This shows that the matrices $W(\mathbf{x})$ exhibit the correct transformation behavior

$$W'(\mathbf{x}) = \Omega(\mathbf{x})W(\mathbf{x})\Omega(\mathbf{x})^\dagger. \tag{120}$$

Channels $a = 1, \ldots, m$ can be introduced by taking direct sums: The multi-channel variable

$$\mathcal{W}(\mathbf{x}) = \left(W^1(\mathbf{x}), \ldots, W^m(\mathbf{x})\right), \tag{121}$$

transforms under $\text{Ad} \oplus \cdots \oplus \text{Ad}$ and represents a data point $s_\mathcal{W}(\mathbf{x}) = [\mathbf{x}, \Omega, \mathcal{W}(\mathbf{x})]$ of the bundle

$$E_{\text{Ad} \oplus \cdots \oplus \text{Ad}} \simeq E_{\text{Ad}} \oplus \cdots \oplus E_{\text{Ad}}. \tag{122}$$

Let us introduce the shorthand notations $n\text{Ad} = \bigoplus_{a=1}^n \text{Ad}$ and $\text{Ad}^n = \bigotimes_{a=1}^n \text{Ad}$.

## 4.3 Equivariant layers

The convolutional layer of Eq. (17) can be viewed as a discretization of a continuous convolution

$$[\psi \star \mathcal{W}]^a(\mathbf{x}) = \sum_b \int_{\mathbb{R}^D} \mathrm{d}\mathbf{y}^D \, \psi^{ab}(\mathbf{y} - \mathbf{x}) U_{\mathbf{x} \to \mathbf{y}} W^b(\mathbf{y}) U_{\mathbf{x} \to \mathbf{y}}^\dagger, \tag{123}$$

with kernel components $\psi^{ab} : \mathbb{R}^D \to \mathbb{R}$ that are non-zero only on the coordinate axes and

$$U_{\mathbf{x} \to \mathbf{y}} = \mathcal{P} \exp\left\{ i \int_0^1 \mathrm{d}s \frac{\mathrm{d}x^\nu(s)}{\mathrm{d}s} A_\nu(x(s)) \right\} \tag{124}$$

is the parallel transporter along the straight line from $\mathbf{x}$ to $\mathbf{y}$.

As discussed in Section 2.3, this design choice is due to parallel transport being path-dependent and the non-uniqueness of shortest paths on the lattice. This convolution is a linear transformation by virtue of being an integral operator, and it maps $\mathcal{W} = (W^1, \ldots, W^m)$ to $\mathcal{W}' = (W'^1, \ldots, W'^n)$ in a gauge equivariant manner:

$$[\psi \star T_\Omega \mathcal{W}]^a(\mathbf{x}) = T_\Omega[\psi \star \mathcal{W}]^a(\mathbf{x}). \tag{125}$$

The action of Eq. (123) on data points $s_\mathcal{W}(\mathbf{x}) = [\mathbf{x}, \Omega, \mathcal{W}(\mathbf{x})]$ is therefore well-defined and independent of the choice of representative for the equivalence class. Thus, the continuous convolution defines a gauge equivariant linear layer

$$\Phi_{\mathrm{Conv}} : \Gamma(m\mathrm{Ad}) \to \Gamma(n\mathrm{Ad}), \qquad s_\mathcal{W} \mapsto s_{\mathcal{W}'}. \tag{126}$$

Analogously, the original bilinear layer in Eq. (22) is a straightforward discretization of

$$W''^a(\mathbf{x}) = \sum_{b=1}^{m} \sum_{c=1}^{m'} \alpha^{abc} W^b(\mathbf{x}) W'^c(\mathbf{x}), \tag{127}$$

and can be linearized using tensor products. If $s_\mathcal{W} \in \Gamma(m\mathrm{Ad})$ and $s_{\mathcal{W}'} \in \Gamma(m'\mathrm{Ad})$, then

$$s_{\mathcal{W} \otimes \mathcal{W}'} \in \Gamma(m\mathrm{Ad} \otimes m'\mathrm{Ad}) = \Gamma(mm'\mathrm{Ad}^2). \tag{128}$$

That is, $\mathcal{W} \otimes \mathcal{W}' = (W^b \otimes W'^c)$. If we let $n$ be the number of output channels, the transformation in Eq. (127) defines a gauge equivariant linear layer

$$\Phi_{\mathrm{Bilin}} : \Gamma(mm'\mathrm{Ad}^2) \to \Gamma(n\mathrm{Ad}), \qquad s_{\mathcal{W} \otimes \mathcal{W}'} \mapsto s_{\mathcal{W}''}. \tag{129}$$

For each channel, trace layers $W'^a(\mathbf{x}) = \mathrm{Tr}(W^a(\mathbf{x}, \Omega))$ compute the trace along the $N \times N$ matrix structure and transform under the trivial representation $\rho = \mathrm{Id}$. Gauge equivariance with respect to the trivial representation is equivalent to gauge invariance, so trace layers are gauge equivariant linear layers

$$\Phi_{\mathrm{Trace}} : \Gamma(m\mathrm{Ad}) \to \Gamma(m\mathrm{Id}), \qquad s_\mathcal{W} \mapsto s_{\mathcal{W}'} \tag{130}$$

Finally, the non-linear activation functions $W'^a(\mathbf{x}) = \nu^a(\mathcal{W}(\mathbf{x})) W^a(\mathbf{x})$ transform $\mathcal{W}(\mathbf{x})$ by scaling each locally transforming variable $W^a(\mathbf{x})$ using a non-linear and gauge invariant function $\nu^a$. Since these activation functions do not affect the transformation behavior of the input feature map $\mathcal{W}$, they are gauge equivariant non-linear layers

$$\Phi_{\mathrm{Act}} : \Gamma(m\mathrm{Ad}) \to \Gamma(m\mathrm{Ad}), \qquad s_\mathcal{W} \mapsto s_{\mathcal{W}'}. \tag{131}$$

### 4.4 L-CNNs for data in other representations

The original L-CNN requires that (input) data take the form of $\mathbb{C}^{N \times N}$-valued matrix variables $W(\mathbf{x})$. However, the connection to the bundle theory for equivariant neural networks makes this requirement relatively straightforward to generalize to functions $f(\mathbf{x})$ taking values in a linear space $V$, and which transform under gauge transformations as

$$f(\mathbf{x}) \mapsto \rho(\Omega(\mathbf{x})) f(\mathbf{x}). \tag{132}$$

This more general L-CNN uses the same trivial principal bundle $P = \mathbb{R}^D \times \mathrm{SU}(N)$ with the same gauge links, only the data is more general. This allows us to also consider matter fields as input: for example, fields in the fundamental representation $\mathbb{C}^N$ (e.g. quark fields) transform according to

$$\rho(\Omega(\mathbf{x})) f(\mathbf{x}) = \Omega(\mathbf{x}) f(\mathbf{x}), \tag{133}$$

where $\Omega(\mathbf{x})$ is a $\mathbb{C}^{N \times N}$ matrix. Similarly, for fields in the adjoint representation (in the sense of adjoint fermions or bosons), we have

$$(\rho(\Omega(\mathbf{x})) f(\mathbf{x}))^a = \Omega(\mathbf{x})^{ab} f(\mathbf{x})^b, \tag{134}$$

with color indices $a, b \in \{1, 2, \ldots, N^2 - 1\}$ and the adjoint matrix

$$\Omega(\mathbf{x})^{ab} = 2\mathrm{Tr}\left[t^a \Omega(\mathbf{x})^\dagger t^b \Omega(\mathbf{x})\right]. \tag{135}$$

In these cases, we have an associated bundle $E_\rho = P \times_\rho V$ consisting of equivalence classes, Eq. (109), and the locally transforming input data $f(\mathbf{x})$ are gauge-dependent representatives of data points $s_f(\mathbf{x}) = [\mathbf{x}, \Omega, f(\mathbf{x})]$. Allowing multiple channels,

$$\mathcal{F} = (f^1, \ldots, f^n), \tag{136}$$

can be accomplished by using direct sum representations $n\rho = \bigoplus_{a=1}^n \rho$.

The convolution in Eq. (123) generalizes to a gauge equivariant layer $\Phi : \Gamma(m\rho) \to \Gamma(n\rho)$ given by

$$[\psi \star \mathcal{F}]^a(\mathbf{x}, \Omega) = \sum_{b=1}^n \int_{\mathbb{R}^D} \mathrm{d}\mathbf{y}^D \, \psi^{ab}(\mathbf{y} - \mathbf{x})\rho(U_{\mathbf{x} \to \mathbf{y}})f^b(\mathbf{y}, \Omega), \tag{137}$$

where $\psi^{ab} : \mathbb{R}^D \to \mathbb{R}$ are kernel components.

Bilinear layers and trace layers do not generalize directly to data that, unlike $W(\mathbf{x})$, are not matrix valued and must instead be tailored to different representations. If $f(\mathbf{x})$ is an $\mathrm{SU}(N)$ vector field, for instance, trace layers could be defined as

$$f'^a(\mathbf{x}) = \mathrm{Tr}\left(f^a(\mathbf{x})f^a(\mathbf{x})^\dagger\right), \tag{138}$$

which acts linearly on $f^a(\mathbf{x}) \otimes f^a(\mathbf{x})^\dagger$ and therefore defines a linear layer $\Gamma(m(\rho \otimes \rho^\dagger)) \to \Gamma(m\mathrm{Id})$. This trace layer is gauge invariant and can be used to define activation functions as gauge equivariant non-linear layers $\Gamma(m\rho) \to \Gamma(m\rho)$ given by

$$f'^a(\mathbf{x}) = \nu^a(w(\mathbf{x}))f^a(\mathbf{x}), \tag{139}$$

for each channel $a = 1, \ldots, m$. Here, $w^a(\mathbf{x}) = \mathrm{Re}\left(\mathrm{Tr}\left(f^a(\mathbf{x})f^a(\mathbf{x})^\dagger\right)\right)$.

### 4.5 The difficulty in achieving full group equivariance

Now that we have connected the original L-CNN to the mathematical theory of equivariant neural networks, it is desirable to do the same for the fully $G$-equivariant L-CNN discussed in Section 3. Fully adhering to the existing theory would require us to identify a suitable principal bundle that describes both the global symmetry under $G$ as well as the local gauge symmetry $\mathrm{SU}(N)$. The complication with this is that gauge symmetry is described by the principal bundle $P = \mathbb{R}^D \times SU(N)$, whereas global symmetry and group equivariance is more closely related to the principal bundle

$$q : G \to \mathbb{R}^D, \qquad q(g) = q(xr) = \mathbf{x}, \tag{140}$$

with structure group $K = G/\mathbb{T}$. It is not obvious whether a single principal bundle can correctly describe both symmetries simultaneously. One interesting candidate is the principal bundle

$$q : G \times SU(N) \to G, \qquad q(g, \Omega) = g. \tag{141}$$

It is of the same general form $q : \mathcal{G} \to \mathcal{G}/\mathcal{K}$ as Eq. (140) but with the total space $\mathcal{G} = G \times SU(N)$ and the structure group $\mathcal{K} = SU(N)$. The corresponding data points are sections of associated bundles $E_\rho = \mathcal{G} \times_\rho V$ and are given by

$$s(g) = [g, \Omega, f(g)] = [g, \mathbb{1}, \rho(\Omega)f(g)], \tag{142}$$

for linear representations $\rho$ of $SU(N)$. Gauge equivariance works similarly as for $P = \mathbb{R}^D \times SU(N)$. However, this bundle describes group equivariance with respect to $\mathcal{G} = G \times SU(N)$, not with respect to $G$ alone. This means that $SU(N)$ would have to represent a global symmetry in addition to the local gauge symmetry. In particular, continuous convolutions would be integrals over $G \times SU(N)$. This is not compatible with the group equivariant L-CNN, so Eq. (141) cannot be the correct principal bundle. The question, then, is whether there is a more appropriate principal bundle or if the bundle theory for equivariant neural networks can be appropriately broadened. We leave this question for future work.

# 5   Conclusions and outlook

In this work, we have reviewed the L-CNN framework (Favoni et al., 2022) from a geometrical perspective and extended the original formulation by accounting for additional global symmetries on the lattice. The L-CNN framework introduced a set of gauge equivariant layers which can be used to build machine learning models for performing computations on gauge link configurations $\{U_{x,\mu}\}$. These layers, consisting of convolutions, bilinear operations, activation functions, and trace layers, are equivariant under lattice gauge transformations. This is achieved by accounting for parallel transport in the definition of gauge equivariant convolutions. In addition, L-CNNs, which are fundamentally based on discrete convolutions on the lattice $\mathbb{Z}^D$, are also equivariant under global translations of the input data.

The global symmetry group $G$ on a hypercubic lattice consists not only of translations, but also includes discrete rotations and reflections. One of the drawbacks of L-CNNs is that they only respect the translational part of the full global symmetry. To remedy this, we have revisited G-CNNs (Cohen & Welling, 2016), which use convolutional layers compatible with general global symmetry transformations. We have first reviewed how to use these $G$-convolutions on vector- and tensor-valued data and then combined the G-CNN approach with the L-CNN framework to obtain network architectures that are not only gauge equivariant, but also equivariant under the full global symmetry group on the lattice. There is a computational drawback associated with this extension: the domain on which feature maps are defined must be enlarged to the full symmetry group $G$, requiring more computational memory. Similarly, it increases the computational complexity of convolutions, as they have to be carried out over feature maps on the group.

Finally, we have linked L-CNNs to the fiber bundle theoretic description of equivariant neural networks (Aronsson, 2022) for the $\mathbb{Z}^D$-equivariant case. This has allowed us to determine the associated bundles used for input data in the original L-CNN formulation and also consider more general input data for different representations of the gauge group. More generally, we have shown how L-CNNs can be understood as a special discretized case of gauge equivariant neural networks on fiber bundles (Gerken et al., 2023). Despite this, a bundle description of $G$-equivariant L-CNNs is still lacking. Identifying the correct principal bundle to simultaneously describe both global $G$ and local SU($N$) symmetry would be a welcome extension of this work.

To conclude, we list some potential applications that could benefit from full $G$-equivariance. Naturally, the most interesting applications are found in four-dimensional SU($N$) lattice gauge theory, which is globally symmetric under discrete translations, rotations, and mirror transformations. Lattice gauge equivariant networks have been used as a preconditioner for the Dirac operator in lattice QCD (Lehner & Wettig, 2023a;b; Knüttel et al., 2024). In these works, gauge equivariant convolution layers include multiple paths and are translationally equivariant but not explicitly symmetric under rotations and reflections. Recently, $\mathbb{Z}^D$-equivariant L-CNNs have been successfully used to learn a classically perfect fixed-point action (Holland et al., 2024), which could help alleviate the problem of large autocorrelation times in Monte Carlo simulations, also known as topological freezing and critical slowing down. Holland et al. (2024) demonstrate that even well-trained $\mathbb{Z}^D$-equivariant L-CNNs violate rotational and mirror symmetry. Machine-learned actions with full $G$-symmetry would be more consistent with the physical theory and likely more accurate as well. Beyond regression problems, $G$-equivariant L-CNNs could find applications in generative models. Equivariant normalizing (Kanwar et al., 2020; Boyda et al., 2021; Albergo et al., 2021b; Abbott et al., 2022) or continuous (de Haan et al., 2021; Gerdes et al., 2023; Bacchio et al., 2023) flow models have been applied to a wide range of lattice field theories in recent years. Flow models can generate statistically independent field configurations and, ideally, exhibit vanishing autocorrelation. In most of these models, lattice symmetries are only approximately realized – in some cases, even translational symmetry is broken down to a subgroup (Boyda et al., 2021). The use of group and gauge equivariant layers would avoid these problems entirely, as the relevant physical symmetries, both global and local, are manifest in the network architecture. Clearly, more work towards an efficient implementation of $G$-equivariant L-CNNs is needed.

### Acknowledgments

The authors thank Andreas Ipp for many helpful discussions regarding group equivariant neural networks and comments on the manuscript. DM and DS have been supported by the Austrian Science Fund FWF

No. P32446, No. P34764 and No. P34455. DM acknowledges additional support from FWF No. P28352. JA has been supported by the Wallenberg AI, Autonomous Systems and Software Program (WASP) funded by the Knut and Alice Wallenberg Foundation. DM and JA thank the organizers of the Banach Center – Oberwolfach Graduate Seminar "Mathematics of Deep Learning", which took place in late 2019 and sparked this collaboration.

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
