# OpenReview forum: "Geometrical aspects of lattice gauge equivariant convolutional neural networks"
_TMLR — Accepted by TMLR_

### Review · Reviewer_UCyp · 2024-02-17

**Summary Of Contributions:**

This paper examines lattice gauge equivariant CNNs and proposes two analyses. First, the authors extend L-CNNs to handle group equivariant constraints. Second, the authors show that L-CNNs (specifically the conv layers) are related to gauge equivariant neural networks (a more general construction) if we recast them using SU(N) bundles.

**Audience:**

No

**Claims And Evidence:**

Yes

**Requested Changes:**

Nothing outside of the weaknesses section.

**Strengths And Weaknesses:**

Strengths
-----------
* The constructions are theoretically principled and explained rather thoroughly.

Weaknesses
--------------
* The material is very hard to approach. I have a background in equivariant neural networks (even touching upon specifically lattice gauge equivariance) and I found this rather hard to parse. Several factors contribute to this, perhaps the largest being the exposition in the form of pure physics (this is specifically Lattice QCD in the broader context of QFT). An exposition closer to math (e.g. dropping the reliance on "Yangs-MIlls theory" "freedom of theory" in Section 2.1) would be immensely helpful in reaching a broader audience.
* Related to the above, but I tend to agree with the original AC comment that this paper would be better suited for a HEP journal (specifically one open to ML constructs). While I could imagine that there is **an** audience (e.g. at least the authors themselves), I struggle to see this fitting into the equivariant ML community or the physics for ML community (which would be perhaps the most relevant areas). This is supported by the fact that the paper builds off work coming from HEP journals (e.g. Favoni et al).
* While theoretically complete, the paper has not demonstrated a practical need for these constructs (in the form of experimental/empirical evidence).

---

> ### Author Response · Authors · 2024-03-08
>
> We thank the Referee for their review. There are several aspects we would like to comment on:
>
> We regret that the Referee believes our work is hard to parse. This is somewhat surprising as we have made the effort to cast the layers proposed in [1] into a language more akin to a mathematically-minded ML audience. For example, section 3 closely follows the conventions and notation used in Cohen and Welling's seminal work on G-CNNs [2], carefully building up the discussion from scalar to tensor-valued feature maps in a rather explicit and, we believe, pedagogical manner. On the other hand, section 4 discusses the bundle theoretic interpretation in a style closely related to recent works in geometric deep learning [3,4]. We admit that section 2 introduces the L-CNN architecture in a style similar to the original work [1], but in our opinion, this should not be seen as a weakness of our work. This is required as a stepping stone if one intends to translate the pure physics approach of the original L-CNN papers for a math and ML audience. If the reader can take the L-CNN layers and the involved symmetries for granted, they are free to skip directly to section 3. It is also perplexing that our work is criticized for providing a primer on Yang-Mills theory. Yang-Mills theory is closely related to all kinds of gauge symmetries, irrespective of the physical or mathematical context.  Gauge symmetries of any kind are intricately linked to so-called connection 1-forms, and Yang-Mills fields are simply what these 1-forms look like after choosing local coordinates and a local gauge. So, whereas our specific formulation of  Yang-Mills theory comes from physics, Yang-Mills theory in general is always relevant whenever gauge symmetry is mentioned. If the Referee is familiar with (lattice) gauge equivariance, but not Yang-Mills theory, then there might also be a misunderstanding involved common to the mathematical ML literature. In the past, works on gauge equivariant networks have often conflated the term gauge equivariance to specifically mean coordinate independence on Riemannian manifolds. While there are strong conceptual similarities, the local symmetries involved in Yang-Mills theories are related to ``internal symmetries'' and gauge equivariance on principal bundles. Our paper could help alleviate this inconsistency between the pure ML and the ML for physics community.
>
> We also disagree that the paper in its current form is not suitable for the TMLR audience due to its reliance on physics. Although the original motivation to study L-CNNs and their extension to group equivariant L-CNNs is of course rooted in physics, our paper is not just targeted at a physics audience. Previous works on L-CNNs were indeed published in physics journals --  as a result, the mathematical structure behind L-CNNs and how they relate to gauge equivariant networks as they are used in the ML community, have largely been neglected. One of the main contributions of our work is to fill in this gap. Our efforts to cast the L-CNNs of [1] into the language of G-CNNs (section 3) and also view them in a bundle theoretic context (section 4), serve as a bridge between the physics and ML community. Both communities share a large interest in the theoretical and practical aspects of symmetry applied to machine learning. The paper is also not about the physics that can be studied with L-CNNs, but the architecture itself.
>
> (1/2)

---

> ### Author Response · Authors · 2024-03-08
>
> We admit that our paper is only theoretical in nature, in the sense that we have not performed any computational experiments or provided an implementation of our group equivariant layers. However, there is numerical evidence that our group-equivariant extension is relevant to physical applications: a recent work applying L-CNNs to particularly difficult regression problems in lattice quantum chromodynamics [5] has demonstrated that even well-trained models appear not to learn discrete symmetries such as rotations and mirror symmetry just from data alone. Rather, these models exhibit deviations from invariance under rotations and flips on the same order of magnitude as the relative prediction error. This suggests that merely shift-equivariant L-CNNs (without full lattice group equivariance) could be limited in their predictive power in actual physics applications. Our proposed layers would entirely solve this issue, but as discussed near the end of section 3, their application, in particular in the four-dimensional case, are prohibitively expensive right now. As such, feasible experiments would likely be restricted to simple toy problems in two dimensions and thus of limited interest. Clearly, more work is required in this regard. Nonetheless, our paper is currently the only proposal for network layers that are compatible with both local/gauge and global symmetry and should therefore still be of interest to the ML/physics community. We will include more details about potential applications in a revised version of the manuscript, which we will upload soon.
>
>
> [1] Favoni et al., Lattice Gauge Equivariant Convolutional Neural Networks, Phys.Rev.Lett., 2022 [arXiv:2012.12901]
>
> [2] Taco S. Cohen and Max Welling, Group Equivariant Convolutional Networks, Proceedings of ICML, 2016, [arXiv:1602.07576]
>
> [3] Gerken et al., Geometric Deep Learning and Equivariant Neural Networks, Artificial Intelligence Review, 2023 [arXiv:2105.13926]
>
> [4] Jimmy Aronsson, Homogeneous vector bundles and G-equivariant convolutional neural networks, Sampling Theory, Signal Processing, and Data Analysis, 2022 [arXiv:2105.05400]
>
> [5] Holland et al., Machine learning a fixed point action for SU(3) gauge theory with a gauge equivariant convolutional neural network, 2024 [arXiv:2401.06481]
>
> (2/2)

---

### Review · Reviewer_6LYn · 2024-02-18

**Summary Of Contributions:**

A number of recent works have looked at exploiting gauge symmetries in neural network design via gauge equivariance. This is a very promising direction of research because gauge groups can get extremely large (introducing degrees of freedom at each point in space), and thus can lead to great improvements in data efficiency. The L-CNN is a translation and SU(N)-gauge equivariant network that has been proven useful for predicting gauge invariant quantities in lattice gauge theory. The main contribution of this paper is to extend the L-CNN to also incorporate global symmetries of the lattice, using ideas from group equivariant CNNs.

Additionally, the paper contains a very well-written introduction to lattice gauge theory using physics notation (whereas most work on GDL has relied on the more abstract bundle formalism). It also explains all the relevant GDL background very well. Finally, in section 4, the contributions are placed in the bundle-theoretic framework used by various other authors in GDL. The authors are clearly experts in this area.

The action editor noted that the paper is very heavy on physics, and asked whether it is appropriate for TMLR. Personally I think it is situated at one of the frontiers of geometric deep learning, and will find interested readers in this community. There are several papers on gauge equivariant neural networks that have several hundred citations already, so although this is somewhat niche it is not micro-niche.

A major gap in the paper is that it only presents the mathematics, but does not provide an implementation or experiments involving the newly proposed neural network layers. It does however provide some cursory discussion of implementation and performance issues. This is unfortunate as the interested reader will certainly be curious about the improvements in data efficiency that can be obtained relative to plain L-CNNs.

Another thing that would be nice to see is a discussion on the connection of this work to the various "universality" results that have been published for linear gauge equivariant layers, e.g. in [1, 2]. Granted, these are for the smooth case, but it seems similar results could be proved for lattices quite easily. One would like to know: are the proposed layers indeed the most general gauge equivariant ones?

At the end of the paper, the authors ask the question "Identifying the correct principal bundle to simultaneously describe both global G and local SU(N) symmetry would be a welcome extension of this work". Although I don't have the answer, I would recommend the book [3] where related issues are discussed.

All things considered, I would recommend the paper for acceptance in TMLR. Having experimental results would greatly strengthen the paper, but if the bar is "some readers will find this interesting", the paper meets the bar as is. It is very well written, technical sound, and has potential applications in lattice gauge theoretic calculations.

[1] Weiler, Maurice and Forré, Patrick and Verlinde, Erik and Welling, Max, Equivariant and Coordinate Independent Convolutional Networks, 2023
[2] Cohen, Taco, Equivariant Convolutional Networks, PhD Thesis, 2021
[3] Rudolph, Gerd and Schmidt, Matthias, Differential Geometry and Mathematical Physics: Part II. Fibre Bundles, Topology and Gauge Fields, 2017

**Audience:**

Yes

**Claims And Evidence:**

Yes

**Requested Changes:**

- It would be nice to see a demonstration of improved data efficiency. However, given how many times this has been demonstrated on various problems, it is sufficiently plausible that we would see meaningful gains that I don't consider this strictly necessary for acceptance.
- A discussion on universality

**Strengths And Weaknesses:**

Strengths:
+ Well-written
+ Novel gauge+globally equivariant layers
+ Nice intro to gauge theory

Weaknesses:
- No experiments

For details, see the main review.

---

> ### Author Response · Authors · 2024-03-08
>
> We thank the Referee for their positive review and for pointing out several interesting aspects that we would like to comment on:
>
> We share the Referee's interest in computational experiments showing the potential benefits of group equivariance in L-CNNs. Naturally, the most interesting applications are to be found in four-dimensional lattice gauge theory. To cite some examples from high-energy physics, gauge equivariant (but not fully group equivariant) layers were used in regression problems [1, 2] and generative models [3, 4]. The violation of lattice symmetries is discussed in some of these papers, e.g. Fig. 15 in [1], and section IV C in [3]. We suspect that the gauge equivariant networks used in these applications are at least in part limited due to broken symmetries. Our proposed layers provide a group equivariant solution to these problems. We will include a discussion of potential applications in a revised version of the manuscript.
>
> Unfortunately, as discussed in section 3.9 of our paper, the main issue with the current formulation of our layers is its memory footprint and computational complexity. On four-dimensional lattices, the stabilizer subgroup K consists of 384 elements. Since our approach requires promoting feature maps to be defined on the full group G, the computational cost of G-convolutions is likely prohibitive. In principle, it would be feasible to perform experiments in two dimensions with only eight elements in K, but such experiments are likely of limited interest.  Additionally, the paper is quite long as it is, so we ultimately decided against conducting experiments. We understand that this is a clear weakness of our work. However, given that there have been no previous proposals for combined global and gauge symmetric layers, we believe that our paper is still valuable and sheds some light on the theoretical and practical problems associated with group + gauge equivariant networks. We also believe that the issues in question can eventually be solved, and that the layers discussed in this work are useful as a stepping stone for achieving computationally more feasible network architectures with all desired symmetry properties.
>
> (1/2)

---

> ### Author Response · Authors · 2024-03-08
>
> We also thank the Referee for pointing out the highly interesting question of whether our layers are universal. In the original work on L-CNNs [4], Favoni et al. demonstrate by explicit construction that the combination of gauge equivariant convolutions and bilinear layers can generate any Wilson loop on the lattice. They argue that together with non-linear activation functions, the L-CNN becomes essentially universal: the main objects of interest in (pure) lattice gauge theory are, after all, non-linear functions of Wilson loops. Indeed, even highly non-linear gauge invariant functions such as the fermion determinant can be expanded as a sum over Wilson loops (see e.g. chapter 5.3 of [6]). Ostensibly, one can perform the same explicit construction using our group equivariant layers. While such arguments may be convincing to most lattice QCD practitioners, they are likely not rigorous enough. We are not aware of a theorem stating that all (or at least a certain class of) gauge invariant functions on the lattice can be approximated by a power series of Wilson loops, even though it seems intuitive that this should be the case. The existence of such a theorem would be required in order for the proof in [4] to be complete. Missing such key definitions about what constitutes universality in the context of lattice gauge theory, it becomes difficult to provide formally correct arguments that go beyond intuitive reasoning.
>
> A more tractable question might then be whether the (G-equivariant) L-Conv layer is the most general gauge equivariant linear map. In this context, this specifically means linearity in the locally transforming $W$ variables, not the link variables $U$. We find that the answer to this question is clearly: no, due to path dependence of parallel transport on the principle bundle. In other words, it is a question of holonomy. The L-Conv layer requires parallel transport for gauge equivariance, but the paths one chooses are not unique and, in fact, data at different points might be compared along more than one path. The most general linear layer should therefore, at least conceptually, be represented as a sum over all paths, of which there are infinitely many even on a finite lattice. Such an impractical "path integral formulation" is also mentioned in (Weiler at al, 2023) and [2]. We stress that, as opposed to gauge equivariant networks tailored to particular manifolds such as the Icosahedral CNN, the connection itself is an input to the L-CNN (in terms of link variables). We cannot choose a mere subset of all paths: all different paths must be accounted for as independent contributions to a general linear layer. More work would be needed to study whether one can make sense of such a sum over paths mathematically. We tentatively conclude that the considered gauge equivariant convolutions are not universal linear layers.
>
> We thank the Referee for their comments and are pleased to include these discussions in a revised version of our manuscript very soon.
>
> [1] Holland et al., Machine learning a fixed point action for SU(3) gauge theory with a gauge equivariant convolutional neural network, 2024 [arXiv:2401.06481]
>
> [2] Christoph Lehner, Tilo Wettig, Gauge-equivariant neural networks as preconditioners in lattice QCD, 2023, [arXiv:2302.05419]
>
> [3] Boyda et al., Sampling using SU(N) gauge equivariant flows, Phys.Rev.D, 2021 [arXiv: 2008.05456]
>
> [4] Kanwar et al., Equivariant flow-based sampling for lattice gauge theory, Phys.Rev.Lett., 2020, [arXiv:2003.06413]
>
> [5] Favoni et al., Lattice Gauge Equivariant Convolutional Neural Networks, Phys.Rev.Lett., 2022 [arXiv:2012.12901]
>
> [6] Christof Gattringer, Christian B. Lang, Quantum Chromodynamics on the Lattice, Springer, 2009 [DOI:10.1007/978-3-642-01850-3]
>
> (2/2)

---

> ### Author Response · Authors · 2024-03-15
>
> Dear Reviewer 6LYn,
>
> In the process of revising our manuscript, we came across the paper by Durhuus [1], which clarifies that the linear span of products of Wilson loops is sufficient to describe continuous gauge invariant functions in lattice gauge theory. As detailed in Favoni et al., the L-CNN architecture is able to generate such products of Wilson loops. Thus, the L-CNN architecture is indeed a universal approximator for gauge invariant functions.
>
> [1] B. Durhuus. On the structure of gauge invariant classical observables in lattice gauge theories. Letters in Mathematical Physics, 4(6):515–522, Nov 1980. ISSN 1573-0530. doi: 10.1007/BF00943439

---

> > ### Comment · Reviewer_6LYn · 2024-03-30
> > **Official comment**
> >
> > Great to see confirmation of universality. I still think the lack of experiments is a weakness, but the paper will be of interest to some in the GDL community. So I would still endorse the paper for acceptance.

---

### Review · Reviewer_dkk1 · 2024-03-01

**Summary Of Contributions:**

This works seeks to extend the Lattice Gauge Equivariant Convolutional Neural Network (L-CNN) framework to respect not only equivariance with respect to global translations of input data, but more general action by the global symmetry group. To accomplish this, the paper revisits G-CNNs (Cohen & Welling, 2016) (which introduced convolutional layers compatible with general global symmetry transformations) and adapts this work to the L-CNN framework. The result is a network architecture that is not only gauge equivariant, but also equivariant under action by the full global symmetry group on the lattice. Generality is also lifted considerably: the paper reviews G-convolutions on vector- and tensor-valued data and extends the aforementioned framework to work in such contexts. Some additional analysis is done to show how convolution in L-CNNs arises as a special case of gauge equivariant neural networks on $\text{SU}(N)$ principal bundles (for the $\mathbb{Z}^D$-equivariant case).

**Audience:**

Yes

**Broader Impact Concerns:**

The paper is of a rather theoretical nature; as such, I have no ethical concerns and do not believe the paper requires a broader impact statement.

**Claims And Evidence:**

Yes

**Requested Changes:**

The writing in this paper is, for the most part, very good. That being said, I have some high-level comments. First of all, Section 4, about how L-CNNs arise as a special case of gauge equivariant neural networks on $\text{SU}(N)$ principal bundles, seems mostly like appendix material to me. Although it is a neat connection, it is not seem central to the overall methodology and is auxiliary. Section 4.5 in particular about the difficulty of achieving full group equivariance is at best speculative. It would be fine to mention that this is future work and point to a reference/stub in the appendix. I do not think the best place for this material is the main paper. Second of all, there need to be some references added for the basic high energy physics material. There was no reference added for e.g. Section 2.1 on Yang-Mills theory. A basic reference like [a] would suffice, but please choose whichever you prefer. Additionally, a lot of notation/conventions are clearly geared towards someone with a high energy physics background, less so for a math background. For example, elements of $\mathfrak{su}(N)$ were introduced as traceless Hermitian matrices, as opposed to traceless skew-Hermitian matrices (e.g. as they are in [b]). Physicists have the convention of multiplying through by $i$ at the Lie algebra level, but this is not so for everyone. So long as this is made explicit, either by reference or in text, I have no issue. Language like "color indices" (page 22) is also very physics-specific; I understand it is standard in quantum chromodynamics, but in the context of this more mathematical paper, perhaps a term like "gauge indices" would be more appropriate.

I also give a non-exhaustive list of minor corrections below:

- In the abstract: "non-Abelian" -> "non-abelian" (the standard convention is to spell abelian with a lower case "a")
- The sentence following equation (30): "$[f \star \psi] : G \rightarrow \mathbb{R}$" -> "$[\psi \star f] : G \rightarrow \mathbb{R}$" (following convention with equation (28))
- In the prelude to equation (84), you likely want $b >0$ not $b \in \mathbb{R}$, since for $b < 0$ the "nonlinear" activation function would be linear

### References

[a] Peskin, Michael E. and Daniel V. Schroeder. “An Introduction To Quantum Field Theory.” (1995).

[b] Takhtajan, Leon A.. “Quantum Mechanics for Mathematicians.” (2008).

**Strengths And Weaknesses:**

## Strengths

1. The paper offers a novel extension of the L-CNN framework that incorporates equivariance to action by the global symmetry group (as opposed to just translations) and presents this extension in a very general setting (including vector- and tensor-valued data). The buildup in Section 3 (Extending L-CNNs to Group Equivariance) is quite rigorous and careful; a total of nine subsections guide the reader carefully through all relevant constructions, building on group equivariant convolutions for scalars on the lattice, vector and tensor fields, later on L-Convs, as the focus shifts to gauge equivariance, and so on, up through the most general forms given at the end of Section 3.4. Generalizations of all layer types in the original L-CNNs paper (Favoni et al., 2022) are given, including bilinear layers, trace layers, activation functions, and pooling layers.

2. In terms of writing, the paper is quite strong. The writing is clear and most necessarily preliminaries are carefully introduced and built upon throughout the duration of the paper.

## Weaknesses

1. Although the paper spends a lot of time constructing a highly generalized version of the original L-CNN framework, no applications or motivating example is mentioned. Is there actually a concrete benefit to be had in existing lattice QFT problems from these generalizations? The answer to this question is very unclear and there is no investigation as to whether each (or any, for that matter) of these steps in generalization is actually necessitated by downstream problems in lattice QFT. The original paper on L-CNNs (Favoni et al., 2022) makes some attempt at demonstrating usefulness of their approach by comparing the predictive power of L-CNNs with regular CNNs in the context of synthetic problems where the task is to predict local, gauge invariant observables. This paper does no such thing nor does it explore concrete contexts in which there would intuitively be considerable benefit over the L-CNN approach. Such an exploration feels like a missing component of the paper. The paper also talking about the increased memory and computational requirements of their method over L-CNNs: there is a multiplicative factor of $|K|$ on both (where $K$ is the stabilizer subgroup in the semi-direct product comprising $G$). However, it is unclear to what extent this $|K|$ is critical and/or prohibitive in real-world applications of this method, since, once again, it is unclear in what contexts this approach yields a significant benefit over e.g. L-CNNs. An investigation of these contexts would also help yield a better understanding of how prohibitive this factor of $|K|$ is and how generally the paper's approach can be applied in practice.

## Verdict

The authors of this work extend the L-CNN framework (Favoni et al., 2022) to enable equivariance with respect to the global symmetry group and do so over very general data (tensor-valued). The paper is well-written and the construction is carefully presented. However, no time is spent considering or discussing contexts in which these generalizations are explicitly useful/necessary, over what was already given in the L-CNN paper. Certainly having a general construction is nice, but are there problems in lattice QFT that benefit considerably from this machinery? This question was left unanswered, yet I believe it is of high importance for such a paper. Although this is the case, I believe the claims made in the submission are well supported, and the findings of the paper would be interesting to some portion of TMLR's audience, specifically those working at the intersection of equivariant machine learning and high energy physics. As such, both criteria for acceptance to TMLR are satisfied; ergo, I recommend acceptance of this paper.

---

> ### Author Response · Authors · 2024-03-08
>
> We thank the Referee for their positive review.
>
> We appreciate the comment about the lack of applications in our paper. This is indeed an oversight on our part since there are in fact many potential use cases. As also mentioned in our response to Referee 6LYn, most applications are found in high-energy physics, particularly in four-dimensional lattice gauge theory.
>
> For example, one might consider an extension of [1], where L-CNNs were successfully applied to learn a classically perfect fixed-point action to sub-percent relative error - thus beating previous models by roughly an order of magnitude. Using such machine-learned actions in place of the Wilson action in Monte Carlo studies could potentially alleviate problems with critical slowing down and topological freezing, which count among the most pressing issues in state-of-the-art lattice simulations. Notably, in [1] it is demonstrated that L-CNN models violate rotational and mirror symmetry (see Fig. 15).  The error due to symmetry breaking is comparable to the error of the regression itself. This could limit their applicability and might suggest that the performance of these models is limited due to missing symmetry properties.  Group equivariant layers could also be used in other regression problems such as preconditioners for the Dirac equation [2], or even generative models such as equivariant normalizing flow models [3, 4]. In particular, [3] includes an analysis of broken discrete symmetries in section IV C. They investigate a symmetrization approach by averaging the model output over rotated and flipped configurations in two dimensions but conclude that such a brute-force method is not computationally feasible. Flow models with built-in global symmetry would not require such an ad-hoc fix. In fact, an example of a group equivariant approach is found in [5], where the authors apply a continuous flow model with the appropriate symmetries to sampling lattice configurations. However, these models use Wilson loops of fixed size and shape, and as such, they are not treated as dynamical - in contrast to the L-CNN approach. Our proposed layers would not be limited in this way, as they are straightforward generalizations of the original L-CNN layers, which have been shown to generate any loop (Favoni et al., 2022). To summarize, applications of gauge equivariant networks such as the L-CNN or similar architectures in lattice field theory are currently either missing group equivariance (likely because a group equivariant formulation was not known until now) or limited in other ways.
>
> Despite these obvious applications of our layers, providing actual computational experiments is currently too costly for us - both in terms of development time and computational resources. As discussed in our paper, the memory and computational footprint of feature maps on the group G and  G-equivariant L-Conv layers increases by the size of the stabilizer subgroup K. In four dimensions, this factor increases the memory requirement for the forward pass alone almost four-hundredfold. However, given that machine learning models applied to physical problems have generally benefited from including more and more symmetries, it stands to reason that achieving full group equivariance is a worthwhile endeavor. We believe that our theoretical paper, even without experiments, is a worthwhile contribution towards this goal. It is also clear, however, that more work is needed before combined group and gauge equivariance can be implemented efficiently in real-world applications.
>
> Regarding section 4, we politely disagree with the Referee's opinion. We believe that translating the arguably physics-heavy notation of (Favoni et al., 2022) into the bundle-theoretic framework used by the more mathematically-minded ML community, is not only non-trivial but also a valuable step to enable collaboration between these disjoint communities. In our opinion, moving section 4 into an appendix does not improve the paper.
>
> Finally, we appreciate the suggestion to include more references to standard physics textbooks on Yang-Mills theory and to highlight when we choose to use physics-specific notation and conventions. We will adapt the requested changes and upload a revised version of the manuscript soon.
>
>
> [1] Holland et al., Machine learning a fixed point action for SU(3) gauge theory with a gauge equivariant convolutional neural network, 2024 [arXiv:2401.06481]
>
> [2] Christoph Lehner, Tilo Wettig, Gauge-equivariant neural networks as preconditioners in lattice QCD, 2023, [arXiv:2302.05419]
>
> [3] Boyda et al., Sampling using SU(N) gauge equivariant flows, Phys.Rev.D, 2021 [arXiv: 2008.05456]
>
> [4] Kanwar et al., Equivariant flow-based sampling for lattice gauge theory, Phys.Rev.Lett., 2020, [arXiv:2003.06413]
>
> [5] Bacchio et al., Learning Trivializing Gradient Flows for Lattice Gauge Theories, Phys.Rev.D, 2023, [arXiv:2212.08469]

---

> ### Comment · Reviewer_dkk1 · 2024-03-27
> **Official Comment**
>
> Thank you for the response. The comments regarding applicability and potential future work regarding efficient implementations in real-world applications are rather useful and I would encourage the authors to include this in the discussion near the end of the paper.
>
> I maintain my original verdict and believe this paper should be accepted.

---

### Decision · Action_Editor_ewtW · 2024-04-27

**Recommendation:** Accept as is

**Comment:**

As I explained above, the general consensus is that this is well written paper that does a good job of providing clear and convincing evidence in its claims.  The biggest and only real concern is whether its advanced and physics-heavy subject is appropriate for the TMLR audience.

In the end, the majority of the reviewers and I myself agree that there is an audience for this paper at TMLR, so I am recommending acceptance.

**Audience:**

The biggest question with this submission is one of audience.  I feel convinced that there is an audience for this paper at TMLR.  Granted, the topic is fairly niche, but there is a growing community of geometrical deep learning and in particular previous work that considers gauge invariance to warrant inclusion in TMLR.  Since we are asked to consider if *some* of the TMLR audience would find this interesting, I have to say that it meets that bar.

**Claims And Evidence:**

The reviewers are in agreement, the paper does a good job backing up its claims with clear and convincing evidence. That is not an issue here.